# SUMO-mediated regulation of H3K4me3 reader SET-26 controls germline development in *C. elegans*

Cátia A. Carvalho[1], Ulrike Bening Abu-Shach[1], Asha Raju[1], Zlata Vershinin[2,3], Dan Levy[2,3], Mike Boxem[4], Limor Broday[1]*

**1** Department of Cell and Developmental Biology, Faculty of Medical and Health Sciences, Tel Aviv University, Tel Aviv, Israel, **2** The Shraga Segal Department of Microbiology, Immunology and Genetics, Ben-Gurion University of the Negev, Be'er-Sheva, Israel, **3** The National Institute for Biotechnology in the Negev, Ben-Gurion University of the Negev, Be'er-Sheva, Israel, **4** Division of Developmental Biology, Institute of Biodynamics and Biocomplexity, Department of Biology, Faculty of Science, Utrecht University, Utrecht, the Netherlands

* broday@tauex.tau.ac.il

**Data Availability Statement:** All relevant data are within the paper and its Supporting Information files.

## Abstract

Sumoylation is a posttranslational modification essential for multiple cellular functions in eukaryotes. ULP-2 is a conserved SUMO protease required for embryonic development in *Caenorhabditis elegans*. Here, we revealed that ULP-2 controls germline development by regulating the PHD-SET domain protein, SET-26. Specifically, loss of ULP-2 results in sterility and a progressive elevation of global protein sumoylation. In the germline of *ulp-2* null mutant, meiosis is arrested at the diplotene stage and the cells in the proximal germline acquire a somatic fate. Germline RNAseq analysis revealed the down-regulation of numerous germline genes in *ulp-2* mutants, whereas somatic gene expression is up-regulated. To determine the key factors that are regulated by ULP-2, we performed a yeast two-hybrid screen and identified the histone methylation reader, SET-26 as a ULP-2 interacting protein. Loss of SET-26 enhanced the sterility of *ulp-2* mutant animals. Consistently, SET-26 is sumoylated and its sumoylation levels are regulated by ULP-2. Moreover, we detected a reduction in H3K4 tri-methylation (H3K4me3) histone levels bound to SET-26 in the *ulp-2* mutant background suggesting a dependence of this histone reader on balanced sumoylation. Finally, a comparative proteomics screen between WT and *ulp-2* loss of activity identified the predicted methyltransferase SET-27 as a ULP-2-dependent SET-26-associated protein. SET-27 knockout genetically interacts with ULP-2 in the germline, but not with SET-26. Taken together, we revealed a SUMO protease/H3K4me3 histone reader axis which is required for the maintenance and regulation of germline development.

## Introduction

SUMO, a small ubiquitin-like modifier, is covalently attached to target proteins [1,2] and is essential for development and viability in eukaryotes [3]. SUMO modification is highly

**Funding:** This work was supported by the ISF (1878/15 and 2122/19 to LB). The funders had no role in study design, data collection and analysis, decision to publish, or preparation of the manuscript.

**Competing interests:** The authors have declared that no competing interests exist.

**Abbreviations:** FDR, false discovery rate; GO, Gene Ontology; HP1, heterochromatin protein 1; ncRNA, noncoding RNA; NGM, Nematode Growth Medium; PHD, plant homeodomain; POI, proteins of interest; RIN, RNA integrity numbe; SD, standard deviation; WT, wild-type; Y2H, yeast two-hybrid.

dynamic and reversible; its deconjugation is mediated by specific SUMO proteases that cleave the isopeptide bond between the SUMO moiety and substrates [4,5]. The important role of SUMO in the regulation of gene expression has been extensively demonstrated [6], and many transcription factors and chromatin regulators have been identified as SUMO target proteins [7,8]. Sumoylation is associated with both transcriptional repression and activation [9]. Post-translational histone modifications, including methylation, are epigenetic marks that generate binding sites for histone reader proteins, which transduce these signals to downstream effectors [10,11]. Methylation of histone H3 on lysine residues K4, K36, and K79 is often linked to transcriptional activation, whereas methylation events on lysine residues K9 and K27 are associated with gene repression [11,12]. It has been demonstrated that the SUMO machinery contributes to the regulation of repressive histone methylation; for example, sumoylation has been shown to be required for Polycomb group protein (PcG) Pc2 recruitment to repressive H3K27me3 marks in mouse embryonic fibroblast cells [13] and for the activity of the PcG-like protein SOP-2 in *Caenorhabditis elegans* [14]. Interestingly, Pc2 itself acts as a SUMO E3 ligase in mammalian cell culture [15]. Down-regulation of sumoylation by knockdown of Ubc9 in embryonic stem cells leads to genome-wide loss of H3K9me3-dependent heterochromatin [16]. In *Drosophila*, deposition of H3K9me3 marks depends on SUMO and the PIAS SUMO E3 ligase Su(var)2–10, which recruits the SetDB1/Wde complex [17]. Many protein domains mediating reading of histone methylation states have been identified [18], several of which are linked to SUMO modifications. One of these is the chromodomain found in PcG proteins and the heterochromatin protein 1 (HP1). Chromodomain proteins play an important role in maintaining repressed chromatin states [19,20]. In *C. elegans*, knockdown of SUMO alters the chromatin-binding pattern of the chromodomain protein MRG-1 [21]. An additional histone methylation reader domain family is the plant homeodomain (PHD) zinc fingers which includes approximately 100 distinct PHD fingers with histone-binding activity [22]. Structural studies revealed that a modified histone H3 tail is extensively associated with the PHD finger domain, providing a high degree of specificity of this domain to the active mark H3K4me3 [23,24]. PHD domains were suggested to function as SUMO E3-ligases [25]. Several histone readers contain more than one domain and are therefore potentially able to receive and translate complex signals. One of them is the mixed lineage leukemia 5 protein (MLL5), which contains both a Su(var)3–9, Enhancer-of-zeste, Trithorax (SET) domain, and a PHD domain [26]. MLL5 binds to H3K4me3 marks through the PHD domain and associates with chromatin downstream of the transcriptional start sites of active genes [27–29]. Based on a similar sequence of their SET domains, MLL5 appears to be the mammalian homolog of yeast SET3/ SET4 paralogs, *Drosophila* UpSET, and the *C. elegans* SET-9/SET-26 paralogs [30,31]. The unique property of the SET domain in these proteins is the lack of histone methyl transferase activity [26,32,33] (and this study). The SET domains of SET3 and UpSET were shown to be associated with histone deacetylases [32,33] and to regulate H3K9me2 levels [34].

*C. elegans* SET-9 and SET-26 are 2 histone readers with a PHD-SET domain that binds to H3K4me3 marks in germline and somatic genes [31]. Even though they are 97% identical in their sequence, they perform different functions. SET-9 and SET-26 synergize to maintain germline function across generations; however, SET-26 plays an additional role in regulating lifespan in a germline-independent manner and in regulating resistance to heat stress [31,35,36]. The *C. elegans* ULP-2 protein occupies the Ulp2-like branch of the SUMO proteases and shares 17% similarity with the catalytic domains of the mammalian SUMO proteases SENP6 and SENP7 [37]. Proteomics studies revealed that SENP6 is associated with chromatin organization complexes [38] and is required to stabilize the centromeric H3 variant CENP-A [39,40]. Here, we revealed a role for the ULP-2 SUMO protease in regulating a reader of the active H3K4me3 histone methylation mark in the germline. We demonstrated that ULP-2

plays a key role in germline development and maintenance of germ cell fate and identified the SET-26 histone reader as a central target for this activity.

## Results

### Increased sumoylation levels in *ulp-2* mutant inversely shapes fertility output and lifespan

We have previously reported that the SUMO protease ULP-2 is required for embryonic epidermal morphogenesis. Closer examination revealed that in addition, loss of ULP-2 leads to a developmental delay and germline sterility. The first generation of homozygous *ulp-2* mutants is viable but produces 63% lethal embryos [37]. The remaining 37% complete embryonic development and develop into sterile adults. The development of this escaper group was highly delayed compared to both wild-type (WT) controls or F1 *ulp-2* homozygous mutant animals (8 to 9 days, Figs 1A and S1). We measured the brood size (unhatched embryos and larvae) of first (F1) and second (F2) generation *ulp-2* mutant animals and found that F1 *ulp-2* mutants already displayed reduced fertility compared with WT animals (approximately 55% reduction: 153 ± 53 versus 277 ± 30 progeny). The second generation of homozygous mutants was almost completely sterile (approximately 99% sterility, 0.2 ± 2 progeny) (Fig 1B). This suggests that there is a progressive reproductive fitness decline in the homozygous generations of *ulp-2* mutant animals.

Since the reproductive system plays a key role in regulating the lifespan of the organism [41,42], we determined whether the lifespan of the *ulp-2* mutants was affected. We found that the median lifespan of WT animals is 16 days, compared with 19 days in F1 *ulp-2(tv380)* and 24 days in F2 *ulp-2(tv380)* animals (Fig 1C and S1 Data). These results reflect a progressive increase in the lifespan from F1 to F2 animals lacking ULP-2 activity which correlates inversely with the progressive loss of reproductive fitness. Since both generations of homozygous animals are genetically identical but differ in the levels of the maternal *ulp-2* that they have received, we reasoned that the phenotypic differences could be associated with an accumulation of SUMO conjugates over time. Indeed, whereas F1 *ulp-2(tv380)* mutant animals contained double the amount of sumoylated substrates compared with WT (mean = 1.6 ± 0.4 versus 0.7 ± 0.3); the F2 *ulp-2(tv380)* mutant animals contained triple the amount of SUMO conjugates (mean = 2.4 ± 0.7) (Fig 1D and 1E). All together, these results suggest that the accumulation of SUMO conjugates in subsequent generations of *ulp-2* mutant animals contributes to the decline in fertility and concomitant increase in lifespan.

### Loss of ULP-2 activity disrupts the transcriptional program in the germline

We hypothesized that abnormal desumoylation of germline proteins caused by ULP-2 loss impairs the overall developmental program of the germline leading to sterility. To reveal the key affected pathways, we performed transcriptomics analysis of isolated germlines of *ulp-2* sterile animals and WT control animals. Morphologically, the germlines of F2 *ulp-2(tv380)* animals have a reduction of approximately 50% in length (269 ± 58 μm) compared with WT germlines (533 ± 73 μm) (S2A Fig). The mutant germlines also harbor 45% fewer nuclei, indicating that ULP-2 loss of function negatively affects the cellular content of the germline (S2B Fig). The transcriptomics analysis revealed that a total of 7,202 genes are differentially expressed in the germlines of F2 *ulp-2(tv380)* relative to WT, with 3,577 genes down-regulated and 3,625 genes up-regulated (Figs 2A and S2C and S1 and S2 Tables). Gene Ontology (GO) analysis revealed that the down-regulated genes in F2 *ulp-2(tv380)* germlines are associated with biological processes essential for germline maintenance—such as cell cycle regulation, P-

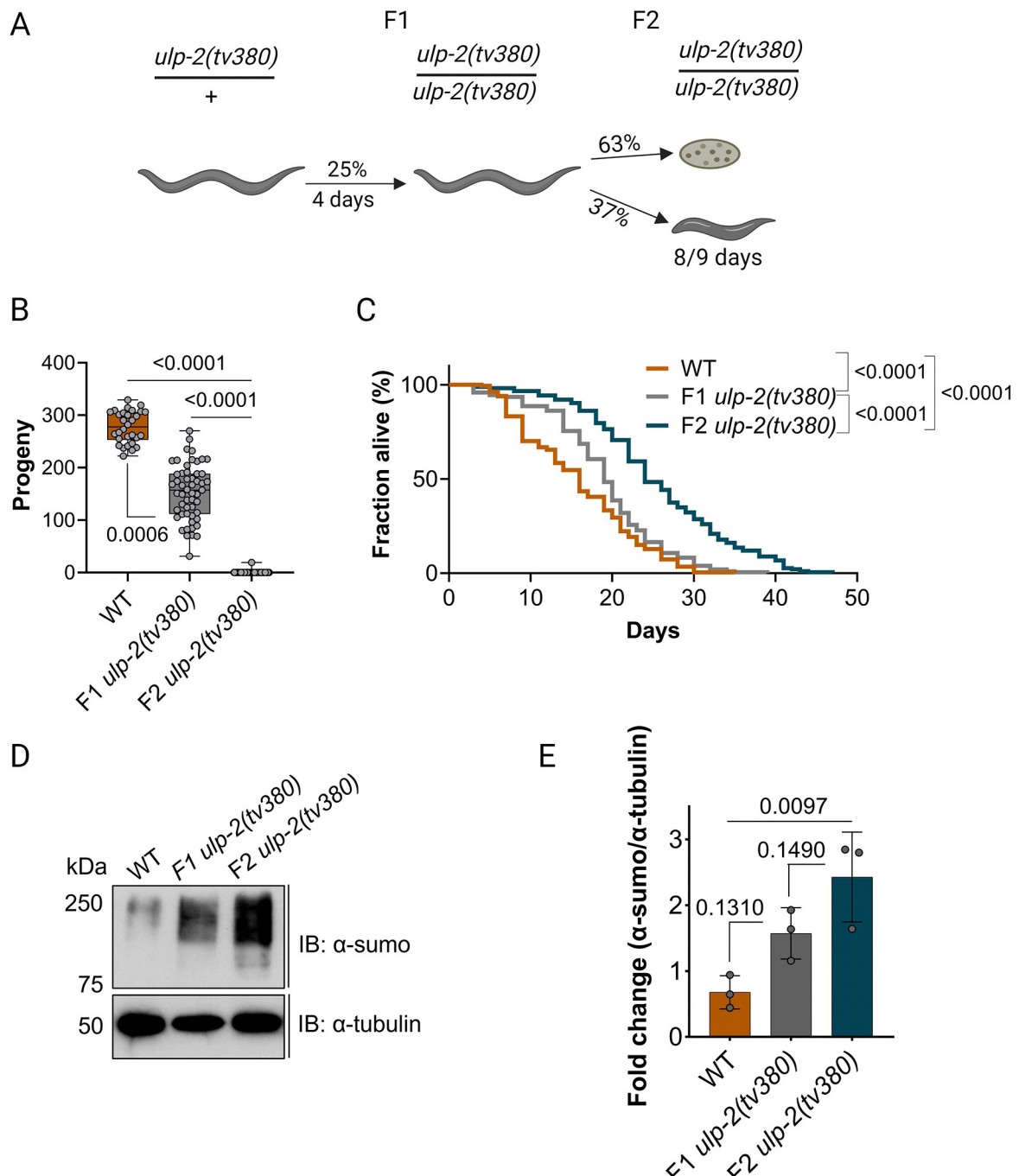

**Fig 1. Loss of function of ULP-2 induces fertility decline and increases longevity accompanied by the accumulation of SUMO conjugates.** (**A**) Schematic representation of the developmental progress of *ulp-2(tv380)* animals. *C. elegans* schemes were generated with BioRender.com. (**B**) Quantification of the number of progeny laid by WT (30 animals), F1 *ulp-2(tv380)* (53 animals), and F2 *ulp-2(tv380)* (94 animals) in 2 biological replicates; the Shapiro–Wilk and one-way ANOVA tests on ranks (Kruskal–Wallis), followed by Dunn's post hoc test, were used; ns = $p > 0.05$. (**C**) Survival curves for WT (252 death events + 100 censored), F1 *ulp-2(tv380)* (220 death events + 162 censored), and F2 *ulp-2(tv380)* (213 death events + 228 censored) animals in 2 biological replicates; the log-rank (Mantel–Cox) test was used; ns = $p > 0.05$. (**D**) Representative immunoblot of the total sumoylation levels of WT, F1 *ulp-2(tv380)*, and F2 *ulp-2(tv380)* animals; 3 biological replicates; the upper panel was probed with anti-sumo and the bottom panel was probed with anti-α-tubulin. (**E**) Quantification of the total levels of sumoylated proteins in D; the Shapiro–Wilk and one-way ANOVA tests, followed by Tukey's post hoc statistical test, were used; ns = $p > 0.05$. The numerical data presented in this figure can be found in S1 Data.

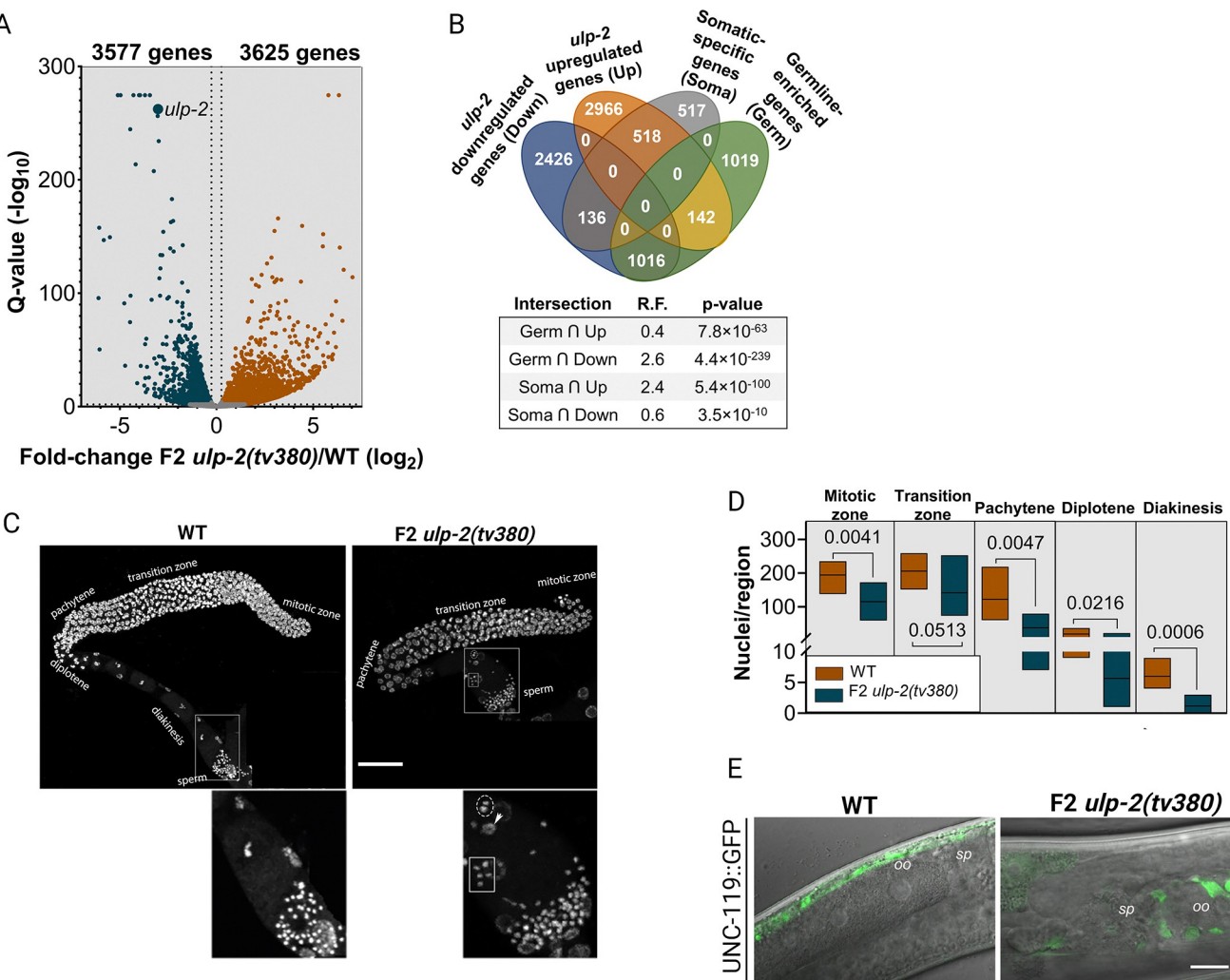

**Fig 2. Loss of ULP-2 disrupts the germline gene expression program, leading to impaired oocyte formation and loss of germ cell identity.** (**A**) A volcano plot displaying the distribution of the differentially expressed genes of the F2 *ulp-2(tv380)* germlines in comparison to WT. The cut-off values used were FDR > 2 (-log10) and fold change>|1.2| (log2). (**B**) A Venn diagram representing the intersection between down-regulated and up-regulated genes in the F2 *ulp-2(tv380)* germlines and the established data sets of germline-enriched and somatic-specific genes [43]; the representation factor (R.F.) of each intersection and the associated *p*-value are shown; Fisher's exact test was used. (**C**) Representative images of the isolated germlines of WT and F2 *ulp-2 (tv380)* animals; the germline zones and sperm are indicated. A single nucleus in diplotene is labeled with dashed circle, a single nucleus in diakinesis is labeled with a rectangle, and a large nucleus is denoted by a white arrow in the *ulp-2(tv380)* germline. Scale bar = 30 μm. Insets below the images show magnification of the proximal gonad and sperm. (**D**) Quantification of the number of nuclei per region in the germlines of WT (7 germlines, 3,834 nuclei) and F2 *ulp-2(tv380)* (6 germlines, 1,809 nuclei) animals; the Shapiro–Wilk and the two-tailed Mann–Whitney U tests were used; ns = *p* > 0.05. Lines on the Y axis indicate a scale change to allow observation of the quantification. (**E**) Representative images of UNC-119::GFP localization in WT and F2 *ulp-2 (tv380)* animals. White line marks the proximal gonad, most proximal oocyte (oo) and spermatheca (sp). Scale bar = 10 μm. The numerical data presented in this figure can be found in S1 Table (A) and S1 Data.

granule assembly, and oocyte development and specification (S3A Fig). On the other hand, the GO analysis of the genes up-regulated in the F2 *ulp-2(tv380)* germlines is associated with somatic processes such as immune response, cuticle development, and neurogenesis (S3B Fig). The intersection of our transcriptomics data set with 2 previously defined data sets [43] substantiates the GO analyses; whereas the down-regulated genes in the F2 *ulp-2(tv380)* germlines largely overlap with the "germline-enriched genes" data set (R.F. = 2.6), the up-regulated genes significantly intersect with the "somatic-specific genes" data set (R.F. = 2.4) (Fig 2B). Taken

together, these observations suggest that ULP-2 promotes the transcription of genes required to maintain the germline and negatively regulates somatic transcription in the germline.

## Impaired desumoylation prevents gamete formation and the maintenance of germ cell identity

We next investigated the cytological defects in the germline that underlie the observed sterility. Germ cells are produced by mitotic divisions in the distal region of the *C. elegans* gonad with meiosis initiating and progressing in the distal to proximal direction. At the proximal end, germ cells mature into oocytes, which are then fertilized in the spermatheca [44]. The gonads of F2 *ulp-2(tv380)* animals are relatively smaller with fewer germ cells and almost no mature oocytes (Fig 2C). The mutant germlines appear to be relatively healthy at the L4 stage but deteriorate when the animals continue development to adulthood. The number of germ cells undergoing mitosis was reduced by 41% in F2 *ulp-2(tv380)* animals. Accordingly, we observed a sharp reduction in the number of nuclei in all meiotic stages. The most dramatic reduction was in diakinesis, with 81% fewer nuclei, resulting in an almost complete absence of oocytes formed in F2 *ulp-2(tv380)* germlines (Fig 2C and 2D). Interestingly, we identified apparently undifferentiated larger misshapen nuclei appearing in variable sizes and numbers in the proximal gonad (Fig 2C, arrow in inset), which resembles the abnormal differentiation of germ cells into somatic cells observed in the *lin-41* mutants [45]. To determine whether these larger cells lost their germline fate, we crossed the *ulp-2* mutant animals with the pan-neuronal marker UNC-119::GFP [46], which has been used to detect germ cell reprogramming to a neuron-like somatic fate [47–49]. We observed germline expression of UNC-119::GFP in cell clusters in the proximal gonad with axonal-like cellular extensions (85%; *n* = 13) that are not detected in WT animals (*n* = 8) (Fig 2E). Analysis of an additional neuronal/hypodermal UNC-13::GFP reporter tagged endogenously with CRISPR/Cas9 also revealed ectopic germline expression in the proximal gonad in F2 *ulp-2(tv380)* mutants (59%; *n* = 32) while no such expression was detected in WT control (*n* = 30) (S4 Fig). This analysis suggests that ULP-2 is essential for protecting germ cell fate in the proximal germline, in line with the results of the transcriptomic analyses.

## ULP-2 interacts with the SET-26 H3K4me3 reader to maintain germline function

To identify targets for the SUMO protease activity of ULP-2 in the germline, we performed a yeast two-hybrid (Y2H) screen of a *C. elegans* mixed-stage cDNA library. The N-terminal region of SENPs/ULPs has been identified as the substrate recognition domain [50,51], and therefore, we used the N-terminal domain of ULP-2 as bait (Fig 3A). We identified the SET-26 protein and a homologous protein, Y73B3A.1, as putative interactors of ULP-2. SET-26 is a PHD-SET protein while Y73B3A.1 has a high sequence homology to SET-26 but lacks the PHD-SET domain (S5A Fig). SET-9 and SET-26 are paralogs that have been established as H3K4me3 readers required for a functional germline [31]; therefore, SET-9, SET-26, and Y73B3A.1 are potential targets for ULP-2 activity in the germline. To examine if *ulp-2* genetically interacts with *set-9*, *set-26*, or *Y73B3A.1*, we first simultaneously down-regulated all 3 genes ("Set3" RNAi) in homozygous F1 *ulp-2(tv380)* animals by RNAi. We observed a decrease in the average number of progeny produced when compared with F1 *ulp-2(tv380)* and WT (S5B Fig). Next, we down-regulated *ulp-2* by RNAi in individual mutants of *set-9*, *set-26*, and *Y73B3A.1*. We observed increased embryonic lethality and near sterility when inactivating *ulp-2* and *set-26*, but not when combining *ulp-2* inactivation with *set-9* or *Y73B3A.1* mutants (S5C and S5D Fig). We therefore decided to focus on the interaction of ULP-2 with SET-26.

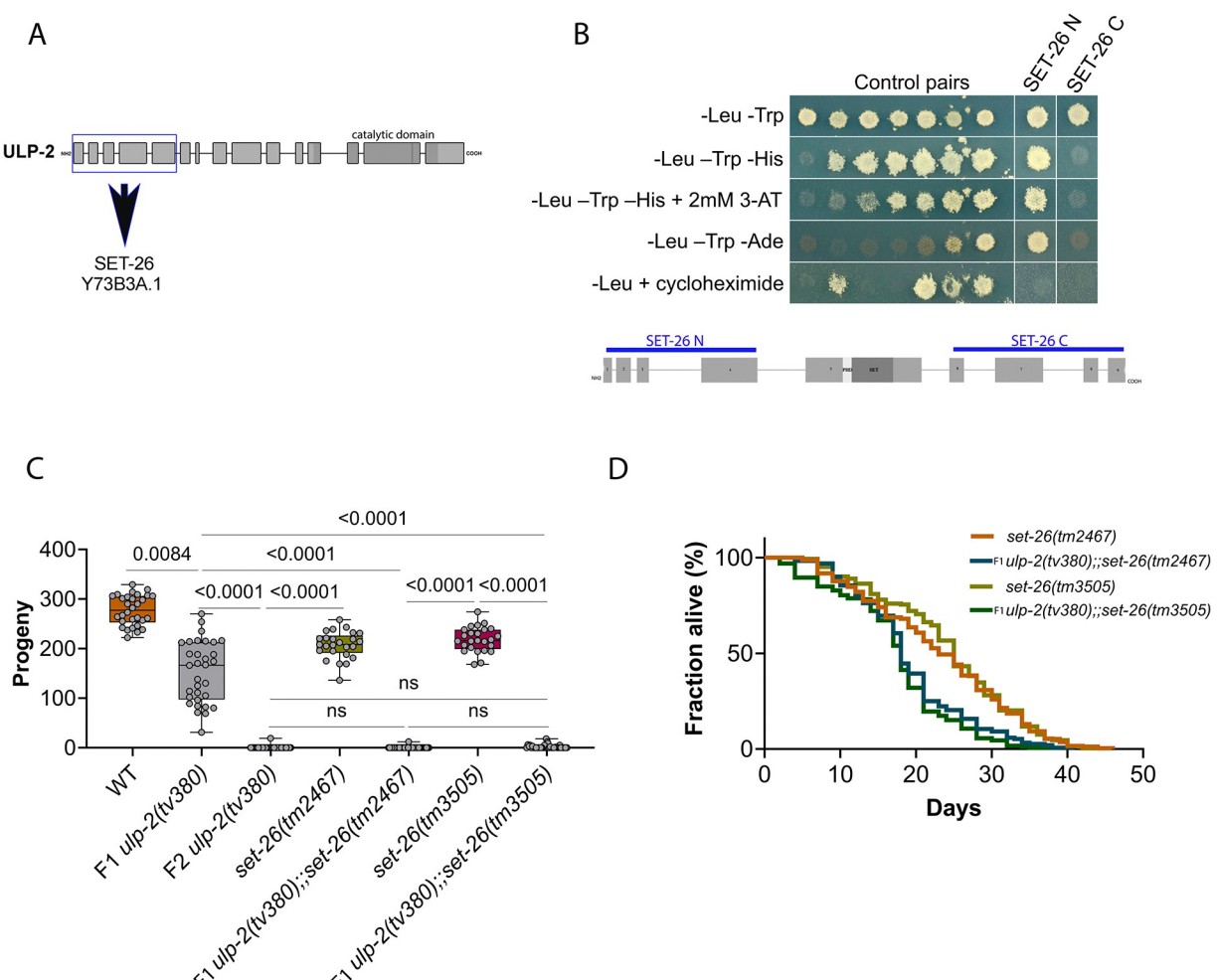

**Fig 3. SET-26 is required for maintenance of the germline function of ULP-2.** (**A**) Exon intron structure of ULP-2 with the region used as bait labeled in blue. (**B**) A yeast two-hybrid analysis of the interaction between the N- and C-terminal domains of SET-26 and the N-terminal domain of ULP-2. The top row is a non-selective growth plate. The middle 3 rows are selective plates that only allow yeast growth in the presence of an interaction. The bottom row is a control plate where growth indicates the auto-activation of reporter genes by the DNA-binding domain protein fusion. Controls are the protein pairs of a known reporter activation strength (S3 Table). (**C**) Quantification of the number of progeny produced by WT (30 animals, 277 ± 30 offspring per mother), *ulp-2(tv380)* (32 animals, 153 ± 63 offspring per mother), F2 *ulp-2 (tv380)* (94 animals, 0.2 ± 2 offspring per mother), *set-26(tm2467)* (25 animals, 207 ± 27 offspring per mother), *ulp-2(tv380);set-26(tm2467)* (39 animals, 0.3 ± 2 offspring per mother), *set-26(tm3505)* (25 animals, 218 ± 25 offspring per mother) and *ulp-2(tv380);set-26(tm3505)* (37 animals, 2 ± 4 offspring per mother); 2 biological replicates; the Shapiro–Wilk and one-way ANOVA tests on ranks (Kruskal–Wallis), followed by Dunn's post hoc test, were used; ns = $p > 0.05$. (**D**) Survival curves for WT (252 death events + 100 censored animals), *ulp-2(tv380)* (220 death events + 162 censored animals), F2 *ulp-2(tv380)* (213 death events + 228 censored animals), *set-26(tm2467)* (254 death events + 92 censored animals), F1 *ulp-2(tv380);set-26(tm2467)* (157 death events + 210 censored animals), *set-26(tm3505)* (245 death events + 81 censored animals), and F1 *ulp-2(tv380);set-26(tm3505)* (205 death events + 188 censored animals); the log-rank (Mantel–Cox) test was used; ns = $p > 0.05$. The numerical data presented in this figure can be found in S1 Data.

To confirm the interaction and map the binding region, we tested the N-terminal (exons 1–4) and the C-terminal (exons 6–9) regions of SET-26 for interaction with ULP-2 by Y2H (Fig 3B). The results indicated that the N-terminal region of SET-26 mediates its interaction with the N-terminal domain of ULP-2 (Fig 3B). To further examine the genetic interaction between ULP-2 and SET-26, we generated double mutants of *ulp-2(tv380)* with 2 loss of function alleles of *set-26*. In these 2 alleles, the N-terminal region is intact and the deletion causes early termination of the reading frame before or inside the PHD domain, leading to a shorter

protein with no PHD and SET domains (S5E Fig). Double *ulp-2; set-26* mutants rendered the first generation of homozygous *ulp-2(tv380)* near sterile, resembling the F2 *ulp-2(tv380)* phenotype (Fig 3C). Thus, F1 homozygous *ulp-2(tv380)* animals express sufficient maternal ULP-2 for germline development, but this process is impaired in the background of the *set-26* loss of function alleles. These results suggest that SET-26 is a key target of ULP-2 activity in the germline.

In addition to its function in the germline, SET-26 has been shown to regulate somatic-driven maintenance of a normal lifespan [35,36]. To determine whether ULP-2 and SET-26 interaction extends to somatic tissues, we measured the lifespan of the double mutants. In double mutants, we observed a shorter extension of lifespan than observed in either single mutant (Fig 3D and S1 Data). This suggests that while in the germline both factors act positively to maintain the germline, in the soma their mode of interaction is possibly indirect and involves additional mechanisms.

## ULP-2 controls the sumoylation levels of SET-26

SUMO proteases mediate the enzymatic removal of SUMO moieties from sumoylated proteins [5]. The physical and genetic interaction between SET-26 and ULP-2 suggests that SET-26 is a target of ULP-2 protease activity. To investigate this hypothesis, we first assessed whether SET-26 is a SUMO-acceptor protein. Using an in vitro sumoylation assay, the SET domain with or without the adjacent PHD domain of SET-26, accumulated multiple higher molecular weight bands detected by anti-SUMO antibody, suggesting that SET-26 can be either sumoylated on several lysine residues or it can be polysumoylated (Fig 4A). The reaction proceeded more rapidly when the region including the PHD-SET domain of SET-26 was included (Fig 4A), suggesting that the PHD domain in SET-26 may act as an intramolecular SUMO E3 ligase, analogous to the enhancement of sumoylation of the bromodomain of the KAP1 corepressor by its SET domain [25]. Next, we performed in vitro desumoylation reactions and found that incubation of sumoylated PHD-SET domains with ULP-2 resulted in the removal of the SUMO peptides, as manifested by a sharp decrease in signal, compared with the sumoylated input (Fig 4B). Overall, these results indicate that ULP-2 can regulate the sumoylation levels of SET-26 in vitro. If SET-26 is a target of ULP-2 in vivo, increased levels of sumoylated SET-26 should be observed in the *ulp-2* loss of function background. Immunoprecipitation of SET-26:: GFP did not exhibit strongly reacting bands with anti-SUMO antibody in a WT background (Fig 4C). However, SET-26::GFP immunoprecipitated from F2 *ulp-2(tv380)* mutant animals exhibited a strong reactivity with the anti-SUMO antibody, corresponding to a 3-fold increase in sumoylation levels (Fig 4C and 4D). These results suggest that ULP-2 regulates the sumoylation levels of SET-26 in *C. elegans*. SET-26 is the ortholog of human MLL5, fly UpSET, as well as yeast SET3 and SET4 that lack histone methyltransferase activity [26,32,33]. We tested whether the sumoylation of the PHD-SET domain of SET-26 can activate the SET methyltransferase in vitro, but no such activity was detected (S6 Fig).

## SET-26 binding to H3K4me3 marks depends on ULP-2

We demonstrated a physical and genetic interaction between ULP-2 and the SET-26 H3K4me3 histone reader. In the context of this histone modification, ULP-2 may act specifically through the regulation of SET-26 or through a direct effect on global levels of H3K4me3. To examine these 2 possible mechanisms, we first tested the role of ULP-2 in maintenance of global H3K4me3 levels. Analysis of H3K4me3 levels in WT and in both F1 and F2 generations of *ulp-2(tv380)* homozygous mutants showed no change in overall H3K4me3 levels (Fig 5A). In the nucleus, SET-26 binds to H3K4me3 marks to regulate gene expression [31]. Hence, we

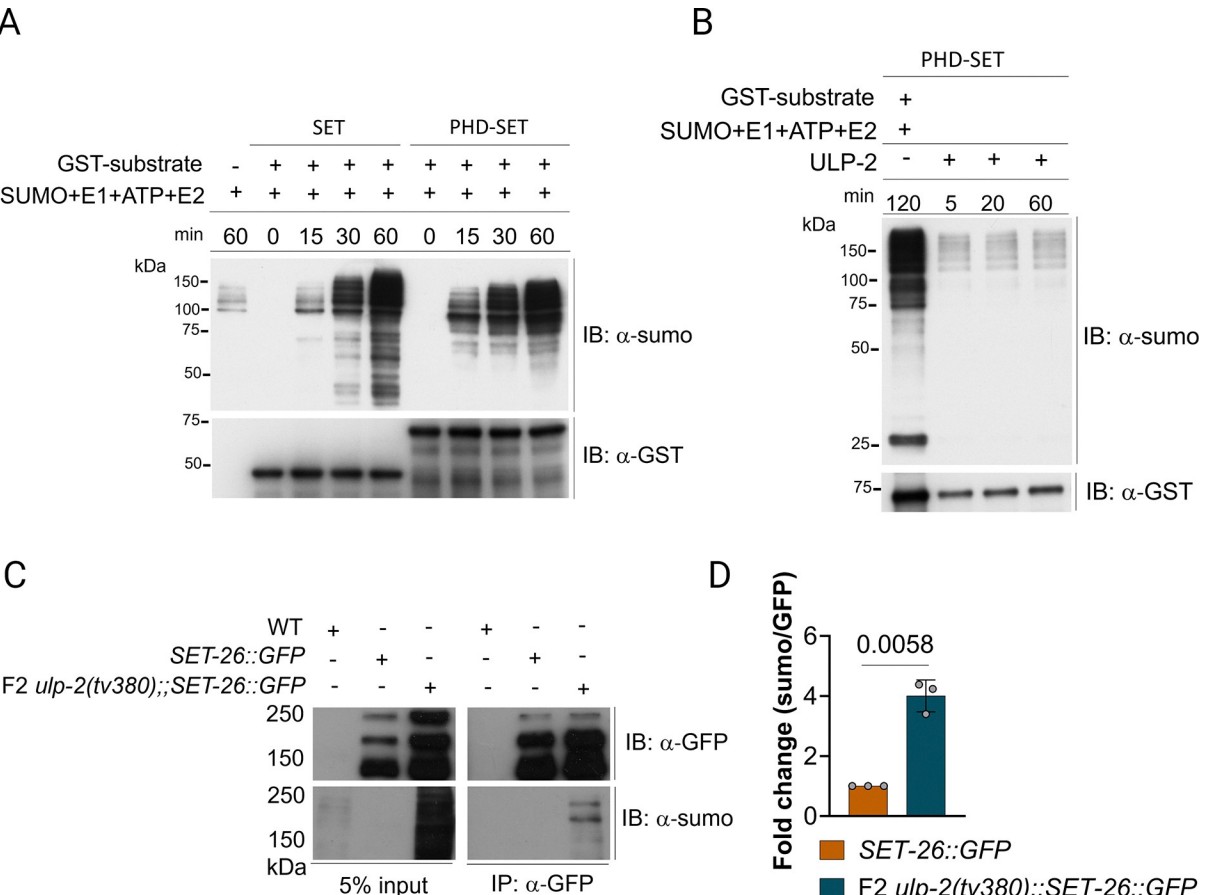

**Fig 4. SET-26 is a SUMO target and is desumoylated by ULP-2.** (**A**) In vitro sumoylation of the SET and PHD-SET domains of SET-26. Bacterially expressed GST-SET and GST-PHD-SET were incubated with E1 (SAE1/SAE2), E2 (UBC9), SUMO1, and ATP for 0, 15, 30, and 60 min at 37°C. Control reactions were incubated for 60 min without the substrate. The anti-SUMO and anti-GST antibodies were used to detect sumoylation and the GST proteins, respectively. (**B**) The ULP-2 catalytic domain desumoylates the PHD-SET domain in vitro. The GST-PHD-SET was sumoylated for 2 h before adding ULP-2 for the indicated time points. Reactions were carried out at 37°C. The anti-SUMO and anti-GST antibodies were used similarly to A. (**C**) SET-26 in vivo sumoylation in WT and F2 *ulp-2(tv380)* backgrounds; anti-GFP and anti-SUMO antibodies were used to probe SET-26::GFP and the sumoylation levels of SET-26, respectively. (**D**) Quantification of the normalized levels of the sumoylation of SET-26 in C for 3 biological replicates; one sample *t* test was used; ns = *p* > 0.05. The numerical data presented in this figure can be found in S1 Data.

intersected a CHIP-seq data set of the SET-9 and SET-26 genomic binding sites (85% SET-26 binding sites) [31] with our transcriptomics data set. Interestingly, although we did not observe a significant intersection with the up-regulated genes (R.F. = 0.8), the overlap between the pool of down-regulated transcripts in the F2 *ulp2(tv380)* mutant germlines and the genes bound by SET-9/26 was significantly overrepresented (R.F. = 2.2) (S7A Fig). This suggests that SET-26 can influence the gene expression program in the germline that we found to be down-regulated in the F2 *ulp2(tv380)* mutant germlines. Thus, we hypothesized that if the SET-26 reader function is altered by an excess of its sumoylation, it can mediate the disruption of the germline gene expression program, leading to the sterility of the F2 *ulp2(tv380)* animals. To test this hypothesis, we first examined if SET-26 nuclear localization in the germline depends on sumoylation. We found that SET-26 localization was not modified following down-regulation of *ulp-2* or the SUMO protein coding gene, *smo-1* (S7B Fig). We next evaluated SET-26 de facto reader ability by measuring its affinity for H3K4me3 marks. Consistent with a previous report [31], immunoprecipitated SET-26::GFP was found to be bound to H3K4me3 marks

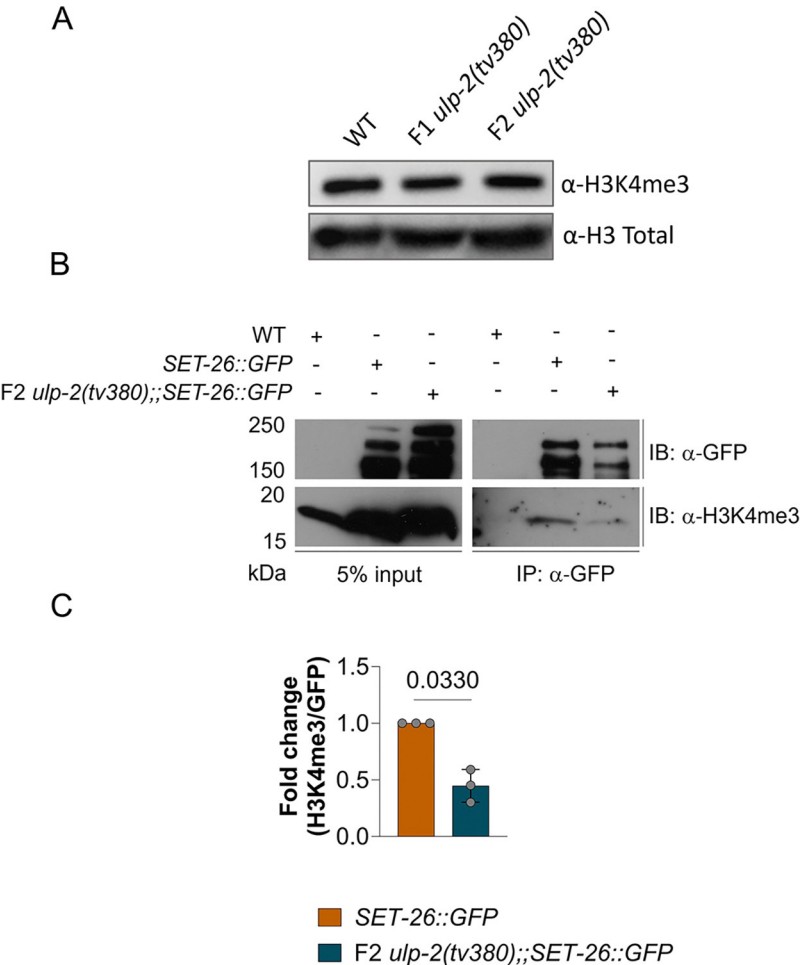

**Fig 5. The sumoylation levels of SET-26 regulate its H3K4me3 reader capacity.** (**A**) Histone H3K4me3 and total histone H3 levels in WT, F1 *ulp-2(tv380)*, and F2 *ulp-2(tv380)* mutant. (**B**) Immunoblot of SET-26::GFP binding to H3K4me3 marks in WT versus F2 *ulp-2tv(380)* backgrounds; anti-GFP and anti-H3K4me3 antibodies were used to recognize the levels of SET-26::GFP and H3K4me3 bound to SET-26::GFP, respectively. (**C**) Quantification of the normalized levels of H3K4me3 binding to immunoprecipitated SET-26::GFP; 3 biological replicates. One-sample *t* test was used; ns = *p* > 0.05. The numerical data presented in C can be found in S1 Data.

in WT (Fig 5B). However, in F2 *ulp-2(tv380)* mutant animals, SET-26::GFP binding to H3K4me3 was reduced by 55% on average, compared with WT (Fig 5B and 5C). Taking into consideration that ULP-2 activity does not alter the global levels of H3K4me3 nor SET-26 nuclear localization, these results indicate that excessive sumoylation of SET-26 in *ulp-2* mutants decreases its H3K4me3 binding affinity, which potentially disrupts the germline gene expression program.

### Excessive sumoylation of SET-26 weakens its protein–protein interactions

Sumoylation can alter the function of a given protein in multiple ways; for instance, it can alter its subcellular localization, stability, or binding with interacting partners [9]. Besides its PhD and SET domains, SET-26 is predicted to be mostly composed of disordered protein regions. These regions assemble multiple protein–protein interactions and are also favored protein regions for SUMO conjugation events [52,53]. To determine whether excessive SET-26 sumoylation could alter its protein interactions, we compared the interactome of SET-26::GFP

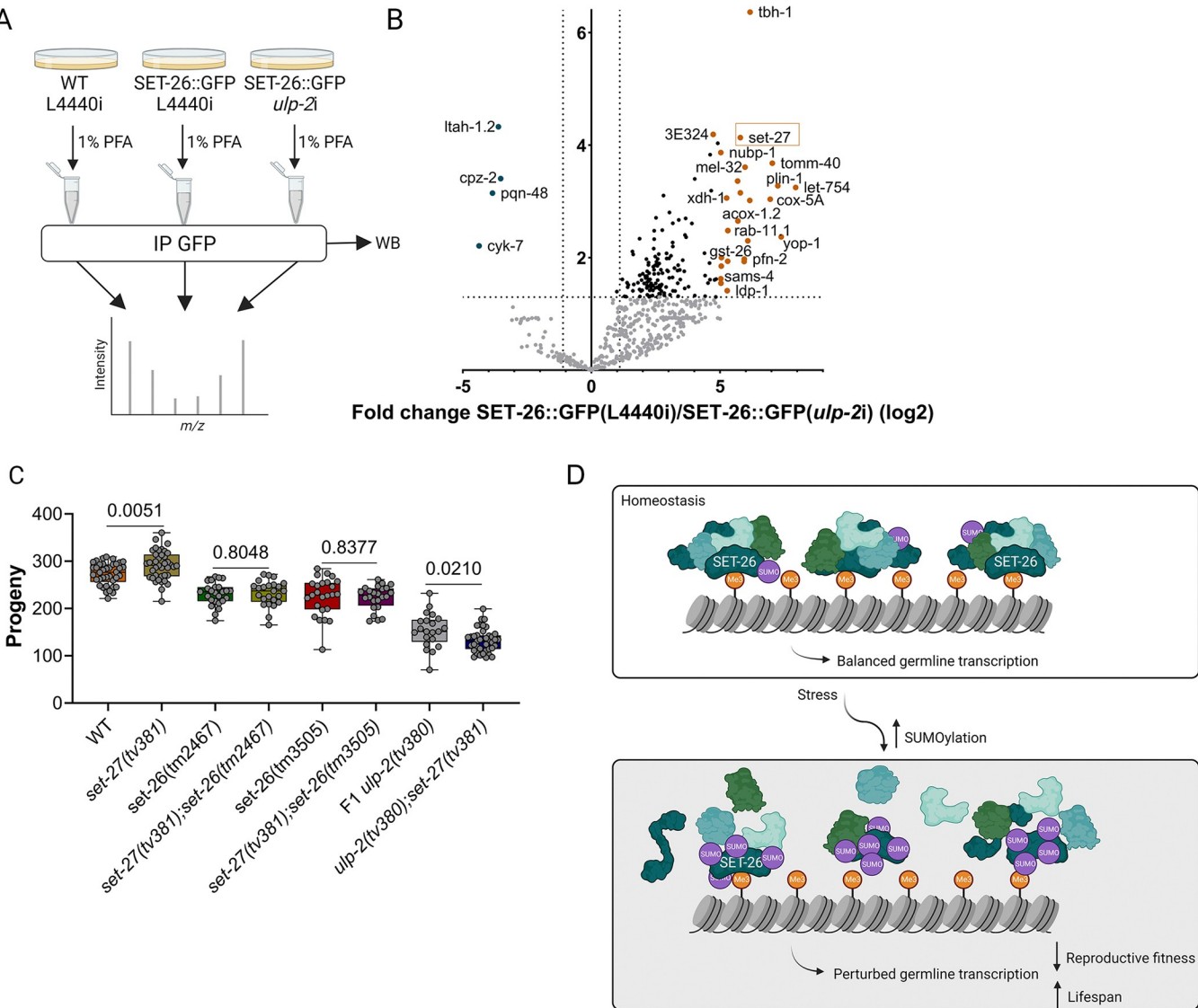

**Fig 6. SUMO-dependent disruption of SET-26 complex formation.** (**A**) Schematic representation of the AP-MS strategy to isolate SET-26::GFP in WT versus an excessive sumoylation background; 3 biological replicates. (**B**) A volcano plot depicting the differentially enriched proteins bound to SET-26::GFP in WT versus an excess of SUMOylation backgrounds; the cut-off values used were FDR > 1.3 (-log10) and fold change>|1.1| (log2). The numerical data presented can be found in S4 Table. (**C**) Quantification of the number of progeny of WT (38 animals, 273 ± 23 offspring per mother), *set-27(tv381)* (39 animals, 291 ± 31 offspring per mother), *set-26(tm2467)* (25 animals, 230 ± 24 offspring per mother), *set-27(tv381);set-26(tm2467)* (24 animals, 232 ± 27 offspring per mother), *set-26(tm3505)* (25 animals, 225 ± 40 offspring per mother), *set-27(tv381);set-26(tm3505)* (24 animals, 223 ± 25 offspring per mother), F1 *ulp-2(tv380)* (21 animals, 153 ± 36 offspring per mother), and F1 *ulp-2(tv380);set-27(tv381)* (39 animals, 132 ± 25 offspring per mother); the Shapiro–Wilk and two-tailed Welch's *t* tests were used; ns = *p* > 0.05; 2 biological replicates. The numerical data presented can be found in S1 Data. (**D**) Model for SET-26 functional dependence on sumoylation. In homeostasis, SET-26 and its interactors bind H3K4me3 marks to regulate gene expression. Unbalanced sumoylation of SET-26 disrupts its reader function by decreasing its binding to H3K4me3 marks. Consequently, germline transcription is perturbed. Generated with BioRender.com.

in the condition of *ulp-2(RNAi)* versus WT through immunoprecipitation and mass spectrometry (Figs 6A, S8A and S8B). In WT, SET-26 was found to be in a complex with several proteins (Fig 6B, right side) whose general biological function concerns metabolic regulation (S8C Fig). However, when SET-26 is excessively sumoylated (*ulp-2(RNAi)*, Fig 6A), we observed a broad decrease in its ability to bind to its WT interacting proteins, and there was only a slight gain in new interactors (Fig 6B, left side and S4 Table). This indicates that excessive

sumoylation of SET-26 decreases its complex-forming capacity, which could directly alter SET-26's ability to either bind to or remain bound to H3K4me3 marks. Of the interaction partners identified, we chose SET-27 for further analysis due to its predicted methyltransferase activity (https://wormbase.org/). To confirm the physical interaction between SET-26 and SET-27, we generated a CRISPR/Cas9-mediated HA-tagged knock in SET-27 strain and crossed it to the SET-26::GFP strain. Immunoprecipitation of SET-27 from the double-tagged strain using the HA epitope showed reactivity with anti-GFP antibody supporting the physical interaction between the 2 proteins (S8D Fig). SET-27 showed no obvious phenotype in an RNAi screen of SET domain proteins [54]. We generated a CRISPR/Cas9-mediated knockout of SET-27 (*set-27(tv381)*) (S8E Fig) to potentially mimic the decreased SET-27/SET-26 interaction observed in the condition of excessive sumoylation (Figs 6B and S8D). We observed that when SET-27 function is lost, there is a small increase of approximately 7% in the number of progeny produced relative to WT, indicating a minor function of SET-27 in the germline (Fig 6C). Double mutant *set-27; set-26* does not impact the number of progeny generated compared with SET-26 loss of function. However, we observed that *ulp-2; set-27* double mutant displayed a reduction of approximately 17% in progeny produced compared with F1 *ulp-2(tv380)* mutants (Fig 6C), suggesting a genetic interaction between SET-27 and ULP-2.

In summary, our study highlighted a positive function of the SUMO protease ULP-2 in the regulation of the SET-26 histone reader protein to enable and maintain the germline transcriptional program (Fig 6D), while SET-27 may play a more minor role in this SUMO axis.

## Discussion

In this study, we revealed that the SUMO protease ULP-2 is required for germline development and for SET-26 function in the germline. The progressive sterility of subsequent *ulp-2* mutant generations was accompanied by increased global sumoylation levels, which could directly result from the non-reversible sumoylation of multiple targets or from a secondary stress response effect that led to a proteome-wide sumoylation [9]. The most dramatic germline defect was a block in meiosis I and specifically the transition from the end of prophase I to metaphase. Moreover, at the proximal region of the gonad where meiosis I progresses to diplotene and diakinesis, we also observed the loss of germ cell fate. This suggests that there is a redundancy with other SUMO proteases in processes regulated by sumoylation during mitosis in the germline as well as during earlier meiotic stages [55]. This redundancy will maintain a balanced sumoylation that is sufficient for germline development and germ cell fate. However, desumoylation processes during the transition from the end of meiotic prophase I may be regulated nonredundantly by ULP-2.

An important point to highlight is that the late arrest phenotype in germline development of the F2 homozygous *ulp-2* mutant animals differs from the complete elimination of H3K27me3 marks in *mes-2/3/6* mutant which causes germline degeneration [56,57] and enhanced developmental plasticity in *C. elegans* embryos [58].

Dramatically, transcriptomics analysis of dissected germlines revealed a large-scale down-regulation of germline genes together with up-regulation of misexpressed somatic genes. The impaired expression of thousands of genes in the germline suggests that knockout of ULP-2 directly affects the chromatin structure in the germline. Moreover, we predict that additional key germline proteins including chromatin factors and transcription factors are regulated by this SUMO protease.

Using a yeast two-hybrid screen, we identified the PHD-SET domain protein, SET-26, as a binding partner of ULP-2. When we compared the CHIP-seq data of SET-9/SET-26 [31] genomic binding sites with our RNAseq *ulp-2* mutant data, we observed that the vast majority of genes that SET-9/SET-26 bind to are down-regulated in *ulp-2(tv380)* mutant animals.

Moreover, double mutant *ulp-2; set-26* resulted in a dramatic enhancement of the sterility phenotype to complete sterility in the first generation of homozygous animals, thus supporting a genetic interaction between *ulp-2* and *set-26*. To elucidate the functional interaction between ULP-2 and SET-26, we investigated whether SET-26 is sumoylated. We found that it is sumoylated on several residues or is polysumoylated, since we observed multiple bands on western blots. In vivo, we observed an increase in SET-26 sumoylation in the *ulp-2* mutant background. Next, we showed that constitutive sumoylation of SET-26 interfered with the direct binding of SET-26 to the H3K4me3 histone marks and its interactors, among them the SET-27 putative methyltransferase. Our study revealed that sumoylation of SET-26 abolished its binding to H3K4me3, which may lead to impairment of the germline transcriptional program. As previously reported [31], the *set-26* mutant alleles increase lifespan; however, in double mutant *ulp-2; set-26* the lifespan extension was lower when compared to single mutants. This suggests that whereas ULP-2 and SET-26 share the regulation of the same germline genes, the two proteins have different chromatin binding sites or transcriptional activities in the chromosomal regions of somatic genes involved in lifespan regulation.

During germline development and maintenance of totipotency, the histone modifiers and histone readers must be tightly regulated. A balance between different histone modifications appears necessary to maintain germline pluripotency [59–62]. In *C. elegans*, the role of SUMO in the epigenetic regulation of the germline was recently demonstrated by the finding that sumoylation of the CCCH zinc-finger PIE-1 and HDAC1 promotes piRNA-dependent transcription silencing and maintains germline fate [63,64]. Another example is the *C. elegans* protein MRG-1, a sumoylated chromodomain protein whose chromatin binding patterns are modified following a decrease in overall sumoylation [21].

Our data suggest that ULP-2 contributes to the formation of an epigenetic barrier that will prevent germ cell reprogramming toward somatic fates using sumoylation-desumoylation cycles. Dynamic sumoylation is an efficient method to strengthen this regulation. Since SET-26 has been shown to bind to and restrict H3K4me3 spread in germline genes [31], it is possible that cycles of sumoylation and desumoylation mediate this binding (Fig 6D).

Several studies have demonstrated the role of PHD domains in recognizing histone methylation [12] and in regulating sumoylation [65]. The PHD domain of the KAP1 corepressor functions as a SUMO-E3 ligase of the adjacent bromodomain [25]. The sumoylation of the bromodomain mediates the direct recruitment of H3K9 HMTase and HDAC activities to promoter regions to silence gene expression [25]. SIZ1 is an *Arabidopsis* SUMO E3 ligase and its PHD domain binds to H3K4me3 [66]. The PHD domain of the SET-26 ortholog, MLL5, has been shown to be recruited to actively transcribed genes and binds specifically to H3K4me3 marks. However, this binding is inhibited by phosphorylation on neighboring residues on the histone (Thr3 or Thr6) [29]. Similarly, as sumoylation modulates protein interactions [8], it is possible that SUMO conjugation masks the interaction surface between the PHD domain of SET-26 and H3K4me3 to regulate the binding of the histone reader to the histone marks.

Histones are also direct substrates of sumoylation. Serial modifications on histones, including H2B ubiquitylation, H3K4 methylation, histone sumoylation, and histone deacetylation, function together to regulate transcription [67]. Histone sumoylation recruits the Set3 histone deacetylase complex to both protein-coding and noncoding RNA (ncRNA) genes [68]. Several SUMO proteases have been shown to regulate histone-modifying enzyme activities during developmental processes. For example, SENP3 regulates the SET1/MLL complex during osteogenic differentiation [69] and the SETD7 histone methyltransferase during sarcomere assembly [70]. In summary, we revealed a unique example of the regulation of a reader of an active histone mark by SUMO. Our study suggests that in addition to the sumoylation of the histones themselves and the sumoylation of histone-modifying enzymes, there is another layer of

chromatin regulation contributed by the balanced sumoylation/desumoylation of the readers of specific histone marks.

## Methods

### *C. elegans* strains and genetics

*C. elegans* strains (Table 1) were cultured according to standard protocols [71]. Maintenance was done using Nematode Growth Medium (NGM) seeded with *E. coli* (OP50) at 20°C.

### Fertility and viability assays

Gravid adults of each genotype were bleached 3 consecutive times prior to the analysis. L4 stage hermaphrodites were individually picked into NGM plates seeded with a drop of OP50. Adults were transferred every 12 h to new plates and the total number of laid eggs was scored. After 24 h, the number of hatched progeny was scored.

### Lifespan assays

Lifespan assays were performed using regular NGM plates seeded with OP50 at 20°C, as previously described [73], but with some alterations. Briefly, gravid adults were bleached 3 consecutive times prior to the assays. Twenty day 1 adult hermaphrodites were transferred to each NGM plate and denoted as "day 1." Worms were scored as "censured" or "dead" every 2 to 3 days and transferred to fresh plates. Plates that became contaminated throughout the experiment were discarded. Worms were scored as "censured" when they crawled out of the plate,

**Table 1. Strains used in this study.**

| Strain name | Genotype | Source |
|---|---|---|
| N2 | wild type | CGC |
| NX399 | *ulp-2(tv380)*/mnl1 | [37] |
| FX30138 | tmC6 (dpy-2(tmls1208)) | [72] |
| NX554 | *ulp-2(tv380)*/FX30138;; otIs45[unc-119::GFP] | This study |
| OH441 | otIs45 [unc-119::GFP] | [46] |
| set-26::gfp (rw25) | *set-26::gfp* | [31] |
| NX537 | *ulp-2(tv380)*/FX30138;; *set-26::gfp* | This study |
| NX441 | *set-26(tm2467)*, outcrossed 5Xs | This study |
| NX462 | *set-26(tm3505)*, outcrossed 5Xs | This study |
| NX463 | *set-9(n4949)*, outcrossed 3Xs | This study |
| FX4083 | *y73b3a.1(tm4083)* | Mitani Lab, National BioResource Project (NBRP) |
| NX579 | *ulp-2(tv380)*/FX30138;; *set-26(tm2467)* | This study |
| NX581 | *ulp-2(tv380)*/FX30138;; *set-26(tm3505)* | This study |
| NX769 | *set-27(tv381)* | This study |
| NX757 | *ulp-2(tv380)*/FX30180; *set-27(tv381)* | This study |
| NX767 | *set-27(tv381)*; *set-26(tm2467)* | This study |
| NX766 | *set-27(tv381)*; *set-26(tm3505)* | This study |
| NX770 | *set-27(tv381)*; *set-26::gfp* | This study |
| NX725 | *set-27::HA* | This study |
| NX724 | *set-27::HA; set-26::gfp* | This study |
| PHX6325 | *unc-13(syb6325[unc-13::SL2::GFP::H2B)* | OH, pers. comm |

exhibited extruded intestine/vulva, or became "bags of worms." Worms were scored as "dead" when their body no longer displayed movement in response to physical touch with a platinum wire and/or started to become transparent.

## Immunoblot analysis of whole-worm lysates

Gravid adults of each genotype were bleached for 3 consecutive rounds before collection. Day 1 adults were washed and collected in M9 buffer and snap frozen in liquid nitrogen. Worm pellets (50 to 100 µl of worms) were thawed on ice and 300 µl of lysis buffer were added (Fig 1D, lysis buffer composition: 50 mM Tris (pH 7.5), 150 mM NaCl, 5 mM EDTA, 1% Triton X-100, 0.1% SDS, 1× Protease inhibitors, 25 mM NEM, 25 mM IAA, and 1 mM PMSF; Fig 5A, lysis buffer composition: 50 mM Tris (pH 7.5), 150 mM NaCl, 5 mM EDTA, 1% Triton X-100, 0.1% SDS, and 1× Protease inhibitors). Pellets were sonicated twice; each cycle was composed of 3× or 5× 5-s pulses (45% power) with 10-s intervals. Lysates were spun down (10,000 rpm, 10 min) at 4˚C and supernatant was collected. Protein concentration was determined using the BCA protein assay kit. Next, 40 µg of protein/sample were mixed with 5× Laemmli buffer and heated at 95˚C for 5 min prior to loading. Proteins were transferred to a nitrocellulose membrane for 90 to 180 min at 4˚C. Membranes were washed with water for 5 min and blocked in 5% milk in 1× TBST for 1 h at room temperature. Primary antibodies were incubated overnight in 5% milk in 1XTBST. Membranes were washed 3× with 1XTBST (10 min each) and subsequently incubated with HRP-conjugated secondary antibody (anti-mouse 1:10,000, 1% milk in TBST, or anti-rabbit 1:10,000, 1% milk in 1× TBST) for 1 h at room temperature. For quantification, peak areas of the bands of proteins of interest (POI) were determined using gel tools in Fiji and normalized to the respective control bands. Fold change was determined by normalizing control samples to 1 and then determining the ratio of normalized POI in treated samples relative to the control samples.

## RNA-seq sample preparation and analysis

Batches of 15 to 20 day 1 adults were transferred to a glass depression slide containing 10 mM levamisole in M9. Worms were cut at the pharynx level with 2 syringe needles. Extruded gonads were separated from the worm body using scalpel blades, collected onto the forcep tip, and then transferred to Eppendorf tubes containing 100 µl of trizol on ice. Up to 100 gonads were collected per 100 µl of trizol, then snap frozen in liquid nitrogen, and kept at −80˚C for less than a week. A total of 250 to 270 gonads/sample were collected for the WT N2 strain and 320 to 340 gonads/sample for the F2 *ulp-2(tv380)* genotype. Total RNA was isolated using the Direct-zol RNA MiniPrep Plus kit following the manufacturer's instructions (ZYMO RESEARCH). The RNA integrity number (RIN) was determined using TapeStation. Next, the Ultra II RNA Library prep kit was used to prepare the mRNA libraries and the HiSeq 2500 system was employed for sequencing. Adapters were trimmed with TrimGalore, read quality was inspected with FASTQC, and reads with less than 30 bp were removed. After trimming, quality reads were mapped to the *C. elegans* reference genome with Tophat2, allowing a maximum of 3 mismatches/read. Uniquely mapped reads (more than 94% of the total reads) were counted using HTseq-count. Gene count normalization and differential expression analysis were performed with DESeq2 with a cut off value at $p = 0.05$ (Benjamini–Hochberg correction). GO analyses were performed using Panther (R. 20221013) [74] with cut off at $p > 0.05$ (FDR correction).

## Microscopy

For Fig 2C, day 1 adults were transferred to a poly-lysine-coated slide in a drop of M9 and levamisole (10 mM). Two syringe needles were used to cut the worm at the pharynx level,

followed by fixation with 1% PFA, then stained with DAPI (10 μg/μl) and mounted in Fluoro-mount-G. Z-stacks (0.3 μm) of the germline were acquired using a Leica TCS SP5 II confocal microscope with a 63× 1.4 oil objective, along with the Leica LAS-AF software. Black boxes were inserted below the images to achieve equal image size and near the proximal gonad in the WT on a nonrelevant DAPI staining of intestinal cells and additional ruptured gonad. The quantifications in Figs 2D, S2A, and S2B were performed using the line tool and the multipoint tool in Fiji software. Confocal and DIC images in Fig 2E were acquired using a Zeiss LSM5 confocal microscope with a 63× 1.4 oil objective. Images for S1 Fig were captured using Nikon eclipse 80i equipped with a DS-Fi3 camera with a 10× objective. Confocal and DIC images in S4 Fig were acquired using Axio Observer 7 Zeiss microscope equipped with a 3i Marianas CSU-W1 spinning disk confocal with a 63×/1.4 Oil M27 objective. Confocal images in S7 Fig were acquired with the same 3i Marianas CSU-W1 system using the SoRa disk for super-resolution.

## Yeast two hybrid

The Gal4-DB::ULP-2 bait construct was generated in vector pMB27, which was derived from vector pPC97 [75] by inserting an oligonucleotide linker that encodes a flexible linker (GGGG) upstream of the cloned ORF, and enables cloning using the AscI and NotI restriction sites. The oligonucleotide linker was ligated into pPC97 digested with SmaI and SacI. The ULP-2 bait fragment used consists of the first 5 coding exons (bp 4-771). The corresponding *ulp-2* sequence was amplified from cDNA by PCR and cloned into pMB27 digested with AscI and NotI.

The Gal4-DB::ULP-2 fusion plasmid was transformed into *S. cerevisiae* strain Y8930 [76], and then mated to a pPC86-Gal4-AD prey library of mixed-stage *C. elegans* cDNAs transformed into *S. cerevisiae* strain Y8800 [76] (a gift from Xiaofeng Xin and Charlie Boone). Yeast expressing putative interacting protein pairs were selected by plating mated yeast on synthetic complete (SC) medium plates lacking leucine, tryptophan, and histidine. At least $2 \times 10^6$ independent colonies were screened. De novo autoactivating yeast colonies were eliminated by a plasmid-shuffling-based counter selection [77]. The identities of candidate interacting proteins, which included SET-26, were then determined by PCR amplification and sequencing of the cDNA inserts.

To confirm the interaction between ULP-2 and SET-26, an N-terminal fragment (residues 1–575, exons 1-4) and a C-terminal fragment of SET-26 (residues 1188–1645, exons 6-9) were cloned into Gal4-AD vector pMB29 [78], transformed into *S. cerevisiae* strain Y8800, mated with strain Y8930 expressing Gal4-DB::ULP-2, and then assayed for protein interaction on selective SC agar plates lacking leucine, tryptophan, and histidine or adenine. Plates lacking histidine were supplemented with 2 mM 3-Amino-1,2,4-triazole (3-AT). In addition, auto-activation was assayed on an SC plate lacking leucine and histidine, supplemented with 1 μg/ml cycloheximide.

Control pairs express various AD/DB fusion pairs (S3 Table) that result in known yeast growth characteristics on the different assay plates, from no growth to strong growth, and act as a benchmark to gauge the strength of Y2H interactions.

## RNA interference (RNAi)

RNAi was performed by feeding as previously described [37]. In brief, HT115 bacteria were transformed with 2 *ulp-2(RNAi)* clones (exons 1-4 and exon 15) or the smo-1, *set-9*, *set-26*, *Y73B3a.1* clones and L4440 control and grown overnight at 37˚C. A minimum of 10 colonies of each RNAi clone were grown in 2 ml LB with ampicillin overnight and diluted 1:100 with

fresh LB+ampicillin medium for an additional 4 to 6 h until OD600 = 1. The RNAi bacterial pellets were concentrated 10 times with M9 and mixed in a 1:1 ratio. Aliquots of 800 μl were seeded in 20 ml NGM plates containing 1 mM IPTG and 25 μg/ml carbenicillin and dried overnight. IPTG (200 μl, 100 mM stock) was spread above the bacterial lawn and allowed to dry before embryos were dropped.

## In vitro sumoylation and desumoylation

The SET domain (AA 945-1107) or the PHD-SET (AA 794-1107) domains of SET-26 were cloned into pGEX-4T1 and transformed into *E. coli* BL21(DE3)pLysS. Colonies were grown for 2.5 h at 37˚C and then induced with 0.1 mM isopropyl β-D-1-thiogalactopyranoside (IPTG) for 4 h at 37˚C. Proteins were purified with lysis buffer (50 mM Tris (pH 7.5), 150 mM NaCl, 0.05% NP-40, EDTA-free proteinase inhibitor, 1 mM PMSF, and 250 μg/ml Lysozyme). Proteins were affinity purified on glutathione resin (GeneScript #L00206) and eluted with 100 mM Tris (pH 8.0) and 10 mM reduced glutathione (Sigma, #G4251). After protein determination, 2.5 μg protein were used for a sumoylation reaction performed according to the manufacturer's protocol (LAE Biotech). In vitro sumoylation reactions were incubated at 37˚C for the indicated time points.

For the in vitro desumoylation reactions, the ULP-2 catalytic domain (AA 501-894) was cloned into pET28a+ and transformed into *E. coli* BL21(DE3). Colonies were grown for 2 h at 37˚C (OD600 ~0.8) and then induced with 1 mM isopropyl β-D-1-thiogalactopyranoside (IPTG) for 4 h at 30˚C. Proteins were purified with lysis buffer (20% Sucrose, 20 mM Tris (pH 8), 1 mM β-mercaptoethanol, 350 mM NaCl, 20 mM Imidazole, 10 μg/ml DNase, 1 mM PMSF, 0.1% Igepal CA630, 20 μg/ml Lysozyme, and EDTA-free proteinase inhibitor). Proteins were affinity purified from lysate by Ni Sepharose 6 Fast Flow resin (GE Healthcare #17-5318-01) and eluted with 20 mM Tris (pH 8), 1 mM β-mercaptoethanol, 350 mM NaCl, and 400 mM Imidazole. Then, they were concentrated using Amicon centrifugal filters 10,000 NMWL. Deconjugation reactions were performed in 25 mM Tris (pH 8), 150 mM NaCl, 0.1% Tween, and 2 mM DTT according to [79], and approximately 5 ng ULP-2. The PHD-SET domain was sumolyated for 2 h before deconjugation at 37˚C for the indicated time points. The anti-sumo antibody 21C7 was used for the in vitro blots (DSHB Cat# SUMO-1 21C7, RRID: AB_2198257).

## In vitro methylation

The methylation assay reactions contained a pre-sumoylated or non-sumoylated GST-tagged PHD-SET domain of SET-26, 2 mCi of 3H-labeled S-adenosyl-methionine (SAM) (Perkin-Elmer, AdoMet), and PKMT buffer (20 mM Tris–HCl (pH 8), 10% glycerol, 20 mM KCl, and 5 mM $MgCl_2$). The reactions were incubated overnight at 30˚C and were then resolved by SDS-PAGE for Coomassie staining (Expedeon InstantBlue) or autoradiography.

## Immunoprecipitation

Samples were mainly collected as described above in "immunoblot analysis" with the following exception: the worm pellets were incubated with 1% of PFA for 15 min at room temperature, washed 2Xs with 50 mM Tris (pH 8) (5 min), and subsequently snap frozen in liquid nitrogen. Samples were thawed on ice and 300 μl of lysis buffer (50 mM Tris (pH 7.5), 150 mM NaCl, 5 mM EDTA, 1% Triton X-100, 0.1% SDS, 1.3× Protease inhibitors, and 25 mM NEM) were added. Samples were sonicated on ice in two-four cycles as indicated above and subsequently spun down at 7,000 rpm, 5 min at 4˚C. Next, a sample of the supernatant were reserved for "input" and the remnant was pre-cleared with 15 to 25 μl of pre-washed Protein G Sepharose

beads (GE Healthcare, 17-0618-01) (3×, 5 min, at 4˚C) for 30 min at 4˚C with rotation. Beads were spun down for 20 s at 12,000 rpm and the supernatant was transferred to an Eppendorf tube containing 8 to 10 μl of anti-GFP antibody (Figs 4C and 5B) or 10 μl anti-HA antibody (S8D Fig) and incubated on ice for 1 h. Next, 30 to 50 μl of pre-washed beads were added and samples were incubated at 4˚C for 1 h with rotation. Antibody-bead conjugates were washed once with lysis buffer and 3× with 50 mM Tris (pH 8) (10 min) (in S8D Fig, the antibody-bead conjugates were washed once with lysis buffer and 2× with 50 mM Tris (pH 8) with 250 mM NaCl). Finally, 5× Laemmli buffer was added and heated at 95˚C for 2 min. The primary antibodies used were anti-GFP (Roche), anti-H3K4me3, anti-sumo antibody 6F2 (DSHB). For the blots of the immunoprecipitation with anti-HA shown in S8D Fig the anti-GFP (Invitrogen) was used (Table 2).

## Label-free mass spectrometry

N2 and SET-26::GFP day 1 adults were bleached and embryos were dropped on 20 cm NGM plates containing IPTG and carbenicillin for RNAi by feeding (L4440 and *ulp-2* clones) as described above. Following RNAi feeding, adult worms were collected and immunoprecipitation with anti-GFP was carried out as described above. Immunoprecipitants were loaded on a 10% acrylamide gel and separated 3 cm within the gel. Gels were stained with Coomassie blue overnight and destained with destaining solution for 3 to 4 h. The proteins in the gel were reduced with 3 mM DTT (at 60˚C for 30 min), modified with 10 mM iodoacetamide in 100 mM ammonium bicarbonate (in the dark, at room temperature for 30 min) and then digested in 10% acetonitrile and 10 mM ammonium bicarbonate with trypsin at a 1:10 enzyme-to-substrate ratio, overnight at 37˚C. The resulting peptides were desalted using C18 tips (homemade stage tips) and were subjected to LC-MS-MS analysis. The peptides were resolved by reverse-phase chromatography on 0.075 × 180-mm fused silica capillaries packed with ReproSil reversed phase material. The peptides were eluted with a linear 60-min gradient of 5% to 28%, a 15-min gradient of 28% to 95%, and a 25-min gradient using 95% acetonitrile with 0.1% formic acid in water at flow rates of 0.15 μl/min. Mass spectrometry was performed using a Q Exactive HF mass spectrometer in a positive mode (m/z 300 to 1,800, resolution 120,000 for MS1, and 15,000 for MS2) using repetitively a full MS scan, followed by collision induced dissociation (HCD, at 27 normalized collision energy) of the 18 most dominant ions (>1 charges) selected from the first MS scan. The AGC settings were $3 \times 10^6$ for the full MS and $1 \times 10^5$ for the MS/MS scans. The intensity threshold for triggering MS/MS analysis was $1 \times 10^4$. A dynamic exclusion list was enabled with an exclusion duration of 20 s. The mass spectrometry data were analyzed using MaxQuant software 1.5.2.8 for peak picking and identification using

**Table 2. Antibodies and dilutions used in this study.**

| Antibodies | |
|---|---|
| **Antibody (dilution)** | **Catalog #, Company** |
| anti-SUMO 6F2 (1:160) | DSHB Cat# SUMO 6F2, RRID:AB_2618393 |
| anti-SUMO-1 21C7 (1:1,000) | DSHB Cat# SUMO-1 21C7, RRID:AB_2198257 |
| anti-α tubulin DM1A (1:1,000) | 3873, CST |
| anti-H3 (1:10,000) | ab1791, Abcam |
| anti-H3 (1:1,000) | ab24834, Abcam |
| anti-H3K4me3 (1:2,000–5,000) | ab8580, Abcam |
| anti-GFP (1:5,000–7,000) | 11814460001, Roche |
| anti-GFP (1: 2,000) | A-11122, Invitrogen |
| anti-GST (B-14) (1:250) | SC-138, Santa Cruz Biotechnology |

the Andromeda search engine, searching against the *Caenorhabditis elegans* proteome from the Uniprot database with a mass tolerance of 6 ppm for the precursor masses and 20 ppm for the fragment ions. Oxidation on methionine and protein N-terminus acetylation were accepted as variable modifications, and carbamidomethyl on cysteine was accepted as static modifications. The minimal peptide length was set to 6 amino acids and a maximum of 2 mis-cleavages was allowed. The data were quantified by label-free analysis using the same software. Peptide- and protein-level false discovery rates (FDRs) were filtered to 1% using the target-decoy strategy. The protein table was filtered to eliminate the identifications from the reverse database, as well as common contaminants and single peptide identifications. The imputation of missing values was set at 20 and peptide intensities were normalized to the peptide's intensities of the SET-26 protein in each sample. Statistical analysis of the identification and quantization results was done using Perseus software (1.6.10.43) [80]. Gene ontology analysis was performed in Panther.

### CRISPR/Cas9 genome editing

Generation of the HA-tagged knock in SET-27 strain and the genome-edited *set-27(tv381)* strain were performed using CRISPR/Cas9 technology adapted from [81]. The injection mix was prepared by mixing 1 μl of Cas9 (IDT), 5 μl of tracrRNA, 1 μl dpy-10 crRNA, 2 μl of crRNA- and incubated at 37˚C for 20 min. Next, 2 μl of *dpy-10* ssODN, 4 μl of repair primer, and 3 μl of ddH2O were added (S5 Table). The mix was spun down for 2 min at 12,000 rpm. Then, 17 μl of the mix were transferred to a new PCR tube and injected into the gonads of N2 day 1 adult worms. P0 animals were transferred individually to new plates after ~12 h. Dpy or roller phenotypes were screened to identify P0 worms with successful Cas9 delivery into their germline. F1 progeny (dpy, roller, or non-dpy) of successfully injected P0 was individually isolated to new plates and allowed to lay for approximately 2 days before genotyping. Genotyping primers are detailed in S5 Table.

### Quantification and statistical analyses

All statistical analysis were performed with GraphPad Prism (version 10.0.2) except for RNA-seq and Mass Spectrometry data as described above. Error and error bars are standard deviation (SD).

### Supporting information

**S1 Fig. Comparison of growth rate between WT and F2 *ulp-2(tv380).* (A)** Representative images of WT and *ulp-2(tv380).* Imaging started upon hatching. Animals were grown in 20˚C and imaged every 12 h. Scale bar = 100 μm. (**B**) Mean length of animals (μm) at each time point (*n* = 10 for each time point and genotype). The numerical data presented in panel B can be found in S1 Data.
(TIF)

**S2 Fig. ULP-2 loss of function reduces the length and cell content in the germline and disrupts gene expression.** (**A**) Quantification of germline length of WT (7 germlines) and F2 *ulp-2(tv380)* (6 germlines); two-tailed Welch's *t* test was used, ns = *p* > 0.05. (**B**) Quantification of the total number of nuclei in WT (3834 nuclei) and F2 *ulp-2(tv380)* (1809 nuclei). Two-tailed Welch's *t* test was used, ns = *p* > 0.05. (**C**) PCA analysis of the RNA-seq data set of WT (3 biological samples, 779 isolated germlines) and F2 *ulp-2(tv380)* (3 biological samples, 990 isolated germlines); pink dots represent the biological replicates of WT isolated germlines and blue dots represent the biological replicates of F2 *ulp-2(tv380)* isolated germlines. The numerical

data presented in this figure can be found in S1 Data and in S6 Table.
(TIF)

**S3 Fig. ULP-2 loss of function induces down-regulation of genes involved in germline development and up-regulation of genes involved in somatic functions in the germline.** (**A**) Gene ontology analysis of the biological processes containing the down-regulated genes in germlines of F2 *ulp-2(tv380)* when compared to WT. Biological processes were cut off at Fold enrichment<4; ns = FDR > 0.05. (**B**) Gene ontology analysis of the biological processes containing the significantly up-regulated genes in F2 *ulp-2(tv380)* when compared to WT. Biological processes were cut off at Fold enrichment<2.5; ns = FDR > 0.05. The numerical data presented in this figure can be found in S2 Table.
(TIF)

**S4 Fig. UNC-13 localization in the proximal germline of WT and *ulp-2(tv380)*.** (**A**) In WT gonads the endogenously tagged UNC-13::GFP reporter is expressed only in the somatic gonad sheath cells (arrow) (*n* = 30). (**B**) Representative image of UNC-13::GFP localization in the germline of F2 *ulp-2(tv380)* animals. A cell expressing UNC-13::GFP in the germline is labeled with a black box. White line marks the proximal gonad, the most proximal oocyte (oo) and spermatheca (sp) or predicted spermatheca (sp*). (*n* = 32). Scale bar = 10 μm.
(TIF)

**S5 Fig. SET-26 interacts genetically with ULP-2 in *C. elegans*.** (**A**) Schematic representation of the exons composing the 3 family members identified in the Y2H screen; SET and PhD domains encoding exons are highlighted in red and yellow, respectively. The sequences including the SET and PHD domains are missing in Y73B3a.1 (dashed blue line). Sequences included in the RNAi vectors for the "Set3" RNAi are labeled with black lines above the scheme of each gene. (**B**) Quantification of brood size of WT (*n* = 10 worms in control; *n* = 19 in "Set3" RNAi) and first generation of *ulp-2(tv380)* mutant animals (*n* = 24 in control; *n* = 23 in "Set3" RNAi); Shapiro–Wilk and one-way ANOVA on ranks (Kruskal–Wallis) followed by Dunn's post hoc test was used, ns = *p* > 0.05. (**C**) Quantification of embryonic lethality of WT (*n* = 8 worms), *ulp-2*(RNAi) (*n* = 32), *set-26(tm3526)* (*n* = 4), *set-26(tm3526);ulp-2(RNAi)* (*n* = 6), *set-9 (n4949)* (*n* = 3), *set-9(n4949);ulp-2(RNAi)* (*n* = 9), Y73B3a.1(tm4083) (*n* = 3), and Y73B3a.1 (tm4083);*ulp-2(RNAi)* (*n* = 10); Shapiro–Wilk and one-way ANOVA on ranks (Kruskal–Wallis) followed by Dunn's post hoc test was used, ns = *p* > 0.05. (**D**) Quantification of the amount of progeny of WT (*n* = 25 worms), *ulp-2(RNAi)* (*n* = 65), *set-26(tm2467)* (*n* = 26), *set-26 (tm2467);ulp-2(RNAi)* (*n* = 80), *set-26(tm3505)* (*n* = 18), and *set-26(tm3505);ulp-2(RNAi)* (*n* = 30); Shapiro–Wilk and one-way ANOVA on ranks (Kruskal–Wallis) followed by Dunn's post hoc test was used, ns = *p* > 0.05. (**E**) Schematic representation of WT and *set-26* deletion alleles (NBRP, Japan) *set-26(tm3526)*, *set-26(tm2467)*, and *set-26(tm3505)*, early stop codon is labeled in red. The numerical data presented in this figure can be found in S1 Data.
(TIF)

**S6 Fig. The PHD-SET domain of SET-26 is enzymatically inactive.** (**A, B**) Recombinant GST-PHD-SET of SET-26 was sumoylated for the indicated time points with or without the addition of E2 (UBC9). Samples were then subjected to an in vitro methylation reaction in the presence of 3H-labeled SAM without (**A**) or with recombinant Histone H3.1 (H3.1) as a substrate (**B**). Samples were subjected to SDS–polyacrylamide gel electrophoresis (PAGE) followed by exposure to autoradiogram as indicated. Human SETD6 served as positive control (PC). Coomassie stain of the recombinant proteins used in the reactions is shown on the bottom.
(TIF)

**S7 Fig. Commonality between the group of down-regulated genes in *ulp-2* mutant germlines and SET-9/SET-26 genomic binding sites.** (**A**) Venn diagram representing intersection between the differential expressed genes in F2 *ulp-2(tv380)* germlines and the SET-9/26 binding genes (CHIP-seq); Fischer's exact test was used, ns = *p* > 0.05, R.F. = Representation factor. (**B**) SET-26::GFP is localized at the nuclear periphery and within the nucleoplasm. Confocal GFP analysis of set-26::GFP in control conditions (L4440 vector, *n* = 7); *ulp-2(RNAi)* (*n* = 7) and *smo-1(RNAi)* (*n* = 7). Scale bar = 10 μm.
(TIF)

**S8 Fig. Down-regulation of ULP-2 weakens SET-26 complex formation.** (**A**) Representative immunoblot of the IP of SET-26::GFP samples sent to Mass Spectrometry; N2 L4440 RNAi was used as a control for GFP immunoprecipitation, SET-26::GFP L4440 RNAi corresponds to SET-26::GFP in a WT background and SET-26::GFP *ulp-2(RNAi)* corresponds to SET-26::GFP IP in the excessive SUMOylation background; 3 biological replicates. (**B**) Representative immunoblot showing excessive SUMOylation in the knockdown of *ulp-2* by *ulp-2(RNAi)* when compared to the control (L4440 RNAi). (**C**) Gene Ontology analysis of the WT interacting partners of SET-26::GFP. Data in S4 Table. (**D**) Co-immunoprecipitation of SET-27::HA with SET-26::GFP. The bands in the anti-GFP blot appear weaker in *ulp-2(RNAi)* immunoprecipitation accompanied by the appearance of additional putative isoform-specific SET-26 band (\*). (**E**) Exon intron structure of SET-27 and *set-27(tv381)* deletion allele. The light gray rectangles and dark rectangles are exons, the dark part of the rectangles is the SET domain, the arrows label the crRNAs binding sites, and the deletion is highlighted.
(TIF)

**S1 Raw Images. Scanned films for Figs 1D, 4A–4C, 5A, 5B, S8A and S8D.**
(PDF)

**S1 Data. The underlying numerical data for Figs 1B, 1C, 1E, 2D, 3C, 3D, 4D, 5A, 5C, 6C, S1, S2A, S2B, S5B, S5C and S5D.**
(XLSX)

**S1 Table. RNAseq data set for *ulp-2(tv380)* vs. WT germlines. S2A Fig.**
(XLSX)

**S2 Table. GO terms for RNAseq *ulp-2(tv380)* vs. WT germlines using PANTHER database. S2B Fig.**
(XLSX)

**S3 Table. Control pairs for Y2H screen. S3B Fig.**
(XLSX)

**S4 Table. Identification and quantification of the proteins in the different experimental groups.** Samples labeled with L are control L4440. Samples labeled with U are *ulp-2(RNAi)*. N2 is the WT strain, SET-26 is the SET-26::GFP strain. The samples that contain SET-26 (3 biological samples with L4440 and 3 biological samples with *ulp-2(RNAi)*) include the protein intensities identified normalized to the quantity of SET-26 in the sample ("Nor"). The mass spectrometry data was analyzed using the MaxQuant software as described in the methods. Statistical analysis of the identification and quantization results was done using the Perseus software. S6B Fig.
(XLSX)

**S5 Table. List of PCR primers, gRNA, and ssOligo donor sequences.**
(XLSX)

**S6 Table. Gene counts normalization of the raw RNAseq data.** Normalization was done using "DESeq2" R package. The WT samples are LBN207, LBN208, LBN209. The *ulp-2(tv380)* samples are LBNX39910, LBNX39911, LBNX39912. S2C Fig.
(CSV)

## Acknowledgments

Several strains were provided by the CGC, which is funded by the NIH Office of Research Infrastructure Programs (P40 OD010440) and by the National Bioresource Project, Tokyo, Japan NBRP, which is funded by the Japanese government. We thank Oliver Hobert (Columbia University, New York, NY) for sending us the UNC-13::GFP strain (PHX6325). The SET-26::GFP strain was sent to us from the Siu Sylvia Lee laboratory at Cornell University, Ithaca, NY. RNA-seq, quality control, and differential expression analyses were conducted by The Technion Genome Center. Proteomics was performed by The Smoler Protein Research Center at the Technion, Haifa, Israel.

## Author Contributions

**Conceptualization:** Cátia A. Carvalho, Limor Broday.

**Formal analysis:** Cátia A. Carvalho, Limor Broday.

**Funding acquisition:** Limor Broday.

**Investigation:** Cátia A. Carvalho, Ulrike Bening Abu-Shach, Asha Raju, Zlata Vershinin, Dan Levy, Mike Boxem, Limor Broday.

**Methodology:** Cátia A. Carvalho, Dan Levy, Mike Boxem, Limor Broday.

**Project administration:** Cátia A. Carvalho, Limor Broday.

**Resources:** Limor Broday.

**Supervision:** Limor Broday.

**Validation:** Limor Broday.

**Visualization:** Cátia A. Carvalho, Limor Broday.

**Writing – original draft:** Cátia A. Carvalho, Dan Levy, Mike Boxem, Limor Broday.

**Writing – review & editing:** Cátia A. Carvalho, Dan Levy, Mike Boxem, Limor Broday.

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
