## [Editor Report · Decision Letter 0]

25 Mar 2024

Dear Dr Broday, 

Thank you for submitting your manuscript entitled "SUMO-mediated regulation of a H3K4me3 reader controls germline development in C. elegans" for consideration as a Research Article by PLOS Biology.

Your manuscript has now been evaluated by the PLOS Biology editorial staff as well as by an academic editor with relevant expertise and I am writing to let you know that we would like to send your submission out for external peer review.

Once your full submission is complete, your paper will undergo a series of checks in preparation for peer review. After your manuscript has passed the checks it will be sent out for review. To provide the metadata for your submission, please Login to Editorial Manager (https://www.editorialmanager.com/pbiology) within two working days, i.e. by Mar 27 2024 11:59PM.

Kind regards,

Ines

--

Ines Alvarez-Garcia, PhD

Senior Editor

PLOS Biology

---

## [Decision Letter · Decision Letter 1]

17 May 2024

Dear Dr Broday,

Thank you for your patience while your manuscript entitled "SUMO-mediated regulation of a H3K4me3 reader controls germline development in C. elegans" was peer-reviewed at PLOS Biology. It has now been evaluated by the PLOS Biology editors, an Academic Editor with relevant expertise, and by two independent reviewers. 

The reviews are attached below. As you will see, the reviewers find the conclusions interesting and significant, but they also raise several concerns that would need to be addressed before we can consider the manuscript for publication. Reviewer 1 thinks you should perform additional experiments to measure growth rates, to check if knocking down SMO-1 can rescue the offspring number phenotype or SUMO accumulation, and to validate the interaction between SET-26 and SET-27. Reviewer 2 would like you to confirm the germline to soma transformation by using endogenous markers and also to rewrite the paper and improve the figures to make it is more accessible for a broad audience.

In light of the reviews, we would like to invite you to revise the work to thoroughly address the reviewers' reports. Given the extent of revision needed, we cannot make a decision about publication until we have seen the revised manuscript and your response to the reviewers' comments. Your revised manuscript is likely to be sent for further evaluation by all or a subset of the reviewers.

**IMPORTANT - SUBMITTING YOUR REVISION**

3. Resubmission Checklist

a) *PLOS Data Policy*

b) *Published Peer Review*

d) *Blurb*

Please also provide a blurb which (if accepted) will be included in our weekly and monthly Electronic Table of Contents, sent out to readers of PLOS Biology, and may be used to promote your article in social media. The blurb should be about 30-40 words long and is subject to editorial changes. It should, without exaggeration, entice people to read your manuscript. It should not be redundant with the title and should not contain acronyms or abbreviations. For examples, view our author guidelines: https://journals.plos.org/plosbiology/s/revising-your-manuscript#loc-blurb

Sincerely,

Ines

--

Ines Alvarez-Garcia, PhD

Senior Editor

PLOS Biology

Reviewers' comments

Rev. 1:

The work focuses on the role of ULP-2, a SUMO protease, in controlling germline development in C. elegans. It highlights ULP-2's regulation of SET-26, a PHD-SET domain protein. The study found that ULP-2 mutant showed increased sterility and altered global protein SUMOylation. RNAseq analysis revealed gene expression changes in ulp-2 mutant gonads, indicating a loss of germline gene expression and increased somatic gene expression. The paper also discussed how SET-26's SUMOylation is regulated by ULP-2 and the impact on H3K4me3 binding. This work presents intriguing discoveries in the post-translational modification of histone readers, exhibits commendable innovation and delves into the mechanisms of epigenetic regulatory mechanisms. Overall, this is an important discovery in the field of epigenetic regulation, yet need major revision before publication. Especially, some areas need refinement, and certain conclusions require further evidence.

Major:

1. Regarding the statement, "These results reflect a progressive increase in the lifespan of animals lacking ULP-2 activity, which correlates inversely with its reproductive function,". An additional experiment measuring growth rates could provide more insights.

2. It would be interesting to see if knocking down SMO-1 can rescue the offspring number phenotype or SUMO accumulation. Exploring whether knocking down SMO-1 changes the protein-protein interaction outcomes dependent on ULP-2 for SET-26 would add depth to the study. Additionally, it would be informative to examine whether and how ulp-2 and SUMO impact the expression and localization of SET-26.

3. Fig 5: The total amount SET-26::GFP is much higher in ulp-2;set-26::gfp than in set-26::gfp, but the IPed set-26::gfp in ulp-2;set-26::gfp is obviously lower than from set-26::gfp sample. Are these representative images. Additionally, the reason to choose H3K4me3, but not include other histone modification markers, should be clarified? What are the genomic loci with changed H3K4me?

4. Fig 6: Additional evidence to support the interaction between SET-26 with SET-27 and its regulatory significance is needed. A second assay, either in vivo or in vitro, could strengthen this claim by confirming that ulp-2 RNAi indeed leads to weakened protein-protein interactions.

Minor:

1. Fig 1D: Please clarify why the molecular weight of sumoylated proteome ranges from 100-250 kD but not lower.

2. Fig 1E: in panel E, can't understand how the signals were quantified and compared. According to panel D, the differences are so obvious.

3. Fig 2C: The labeling in this panel is not clear. Please revise for better clarity.

4. Fig 2E: Does soma no longer express unc-119::GFP in F2 ulp-2(tv380)? This needs to be addressed.

5. Fig 2B: It is very likely that ulp-2 primarily affects the proteome, thereby indirectly impacting the transcriptome? Therefore, including a transcriptome assay does not necessary reflect the function of ulp-2 in vivo.

6. Fig 2G: The reasoning for choosing H3K27me3 over other markers like H3K9me3 is not apparent. Also, a quantification of the Western Blot results is suggested.

7. Fig 3A: What are the control pairs in the Y2H assay? Are they the same two proteins or different known interactions? If same two proteins, how reproducible is this assay. The range of set-26 N & C should be indicated within a schematic domain structure of SET-26.

8. Fig 3B: are tm2467 or tm3505 nulls? Is the interaction domain between set-26 & ulp-2 important? Does set-26 show progressive sterility as ulp-2?

9. Fig 3C: It is very hard to recognize the genotype and lifespan based on so many colors. Need include an additional column chart to compare the media lifespan.

10. Fig 4A: Why does the control lane show a SUMO signal without substrate?

11. Fig. 4B: Consider reducing the amount of ULP-2 if the reaction is not in the linear range.

12. Fig. S1C, S3A, and S3F need clearer labeling and visualization.

13. Fig. S4: the PC lane has no substrate H3.1, why show up? The two panels should on the same gel. It is an in vitro experiment with bacterial expressed proteins, it is hard to tell in vivo situations, for example, need a cofactor. Or the bacteria expressed protein is miss folded?

14. Fig. S5D: SET-26 is very likely enriched in the nucleus, but these classes of protein are very likely enriched in cytoplasm?

Rev. 2:

This manuscript discovers a role for the sumo protease ULP-2 in germline development and potential cofactors. The authors i) define the germline and longevity phenotypes of ulp-2 mutant adults; and ii) establish a physical and functional relationship between ULP-2 and SET-26/MLL5/UpSET. They show that SET-26 is overly sumoylated in ulp-2 mutants and this correlates with weak complex formation with its binding partners and H3K4me association. The implication is that these interactions are critical to repress somatic genes and enhance germline expression in the germ line. The authors have undertaken a range of methods to analyze ulp-2 and define its germ line role.

General comment

The paper would be improved and probably garner more readership, if the authors could give more biological context to their findings in the Abstract, Introduction and Discussion. In the Results, consider opening paragraphs by posing a biological question or hypothesis, rather than citing a phenotype. Currently, the paper is a laundry list of phenotypes and factors (factors with similar names no less!), which is confusing to all but the most devoted C. elegans chromatin jocks. In addition, readers often have an easier time with a conclusion like gene X does a, b, c instead of loss of gene x leads to x, y z or loss of a, b, c. In the Abstract and Results, most of the description is loss of function phenotypes, which may lose people with less genetics than the authors. A summary sentence explaining the final model would be helpful. Along the same line, cartoons added to figures in the main results may help readers understand the nature of mutations, hypotheses, experimental plans etc.

Major Points

Figure 1C and Fig 3C appears to show the same data for some specific genotypes. Eliminate one of these panels or use another independent dataset to represent independent trials in the two figures.

Figure 2E. Could the authors please confirm the germ line to soma transformation using one or more endogenous markers. The days of transgenes, which can show faulty expression, is drawing to a close. Crispr alleles, smFISH, antibodies to endogenous proteins. Any of these would be more convincing, and at least Crispr and smFISH are technically easy to do.

Figure 2G. To what extent is the loss of H3K27me3 driving the phenotype of the enhanced sumoylation? Can these phenomena be separated? Similarly, the role of SET-26 binding to H3K4me3 as shown in figure 5, is this a H3K27me effect? Additional experiments would be welcome, but if not, at least talk about it in the Discussion.

"Synergy” in genetic interactions can be hard to interpret. No additivity can mean two genes function in the same genetic pathway whereas additivity can mean independent pathways or roles. This is an issue for the genetics of ulp-2 with the SET domain proteins. If SET-26 is in the same pathway, why does it synergize? Perhaps one stronger conclusion is that both factors act positively ie in the same direction, for viability and fertility. I suggest the authors rework their interpretations here. They cannot rule out a function in the soma, for example.

Minor points

Is the difference between F1 animals and F2 a maternal contribution. Please test or comment on the possibility that ULP-2 is partially maternally rescued.

Fig 2A. RNAseq fold change of 1.2 is a low threshold for gene changes, especially with such a strong increase in sumoylation. Please use a more stringent cut-off or explain why such a lenient threshold was necessary.

Fig 2C. These images, and the arrows indicating abnormal sperm, are two small to see. Please enlarge them or include zoom in panels that makes clear the phenomenon described.

Fig2D. There are gaps in the boxplots for pachytene and diplotene. These gaps are not explained. Please address what these gaps in the data represent.

Fig 3C. Please comment on the suppression of set-26 mutant longevity seen in the double mutants. Are they 'ulp-2 F1 animals'. If so, please include the 'F2s' in this plot as well or address the genetic relationship in an independent manner.

Fig S3B/C. These data are not interpretable because the control WT+ "SET-3" RNAi is missing. Please perform this control and add it to your figures to help the reader interpret the relationship between these set genes and fertility.

Fig S3A and F. These graphs are unreadable. Please increase font size eg. similar to your axes in S3B/C

Errors and error bars in the text and figures need to be clarified. Are they standard deviation or standard error?

Fig legend 3. Provide a list of the 'control pairs'.

Yeast 2 hybrid experiment: please address in the text the rationale for using the N terminal domain of ULP-2 rather than the full-length or C terminus.

Introduction: "whereas methylation events on histone H3 on lysine residues K9, K20, and K27 are associated with gene repression [11]." This is a typo, I think you mean H3K23me, or do you mean H4K20me?

"although we did not observe a meaningful intersection with the upregulated genes (R.F.=8)," Please define what meaningful means here (explain RF number).

Among genes and proteins found in the screens, is there anything to explain the very specific meiosis phenotype?

Fig 5A. Indicate in the figure and legend that the protein source for the immunoblot is an IP of SET-26::GFP.

Fig 6. Move model from supplement to here.

Note that H3K27me also has a role for embryonic plasticity (Yuzyuk, Dev Cell, 2009).

Fig S2A. Please explain what the different colors mean (red vs orange) and how that differs to blue in S2B.

Fig S3A. Add the region of RNAi homology as a bar over the three genes (or equivalent)

Fig S4 lacks A and B designation of the two panels. Please add.

---

## [Decision Letter · Decision Letter 2]

25 Nov 2024

Dear Dr Broday,

Thank you for your patience while we considered your revised manuscript entitled "SUMO-mediated regulation of a H3K4me3 reader controls germline development in C. elegans" for publication as a Research Article at PLOS Biology. This revised version of your manuscript has been evaluated by the PLOS Biology editors, the Academic Editor and the two original reviewers.

Based on the reviews (attached below), we are likely to accept this manuscript for publication, provided you satisfactorily address the data and other policy-related requests stated below. We will leave up to you if you want to follow Reviewer 2' suggestion.

In addition, we would like you to consider a suggestion to improve the title:

"SUMO-mediated regulation of H3K4me3 reader SET-26 controls germline development in C. elegans"

We expect to receive your revised manuscript within two weeks. 

*Published Peer Review History*

*Press*

Sincerely,

Ines

--

Ines Alvarez-Garcia, PhD

Senior Editor

PLOS Biology

DATA POLICY:

Thank you very much for providing the data underlying the graphs shown in the figure. I have checked all the data and I am missing some, thus we would be grateful if you could provide also the data underlying the graphs shown in the following:

Fig. 2A, Fig. 6B and Fig. S2C

Please also mention where the data can be found in the supplementary figure legends. It seems this information is missing.

CODE POLICY

Reviewers' comments

Rev. 1: Shouhong Guang - note that this reviewer has signed his review.

The authors have done a fabulous work on revision. I completely support its publication in current version.

Rev. 2:

The paper is improved with the new data and writing. I have only one small comment: I would suggest adding the endogenously tagged crispr strain (unc-13 I think it was) to the main figures, and out of the supplement, as it is much more convincing. I have no other objections, the paper is interesting and good.

---

## [Editor Report · Decision Letter 3]

11 Dec 2024

Dear Dr Broday,

Thank you for the submission of your revised Research Article entitled "SUMO-mediated regulation of H3K4me3 reader SET-26 controls germline development in C. elegans" for publication in PLOS Biology. On behalf of my colleagues and the Academic Editor, René Ketting, I am delighted to let you know that we can in principle accept your manuscript for publication, provided you address any remaining formatting and reporting issues. These will be detailed in an email you should receive within 2-3 business days from our colleagues in the journal operations team; no action is required from you until then. Please note that we will not be able to formally accept your manuscript and schedule it for publication until you have completed any requested changes.

PRESS

Sincerely, 

Ines

--

Ines Alvarez-Garcia, PhD

Senior Editor

PLOS Biology
